# G-RepsNet: A Lightweight Construction of Equivariant Networks for Arbitrary Matrix Groups

**Sourya Basu**[*]                                                             *basusourya@gmail.com*
*AWS AI Labs*

**Suhas Lohit**                                                                 *slohit@merl.com*
*Mitsubishi Electric Research Laboratories*

**Matthew Brand**                                                              *brand@merl.com*
*Mitsubishi Electric Research Laboratories*

**Reviewed on OpenReview:** *https://openreview.net/forum?id=k1eYngOvf0*

## Abstract

Group equivariance is a strong inductive bias useful in a wide range of deep learning tasks. However, constructing efficient equivariant networks for general groups and domains is difficult. Recent work by Finzi et al. (2021b) directly solves the equivariance constraint for arbitrary matrix groups to obtain equivariant MLPs (EMLPs), but this method does not scale well and scaling is crucial in deep learning. Here, we introduce Group Representation Networks (G-RepsNets), a lightweight equivariant network for arbitrary matrix groups with features represented using tensor polynomials. The key insight in our design is that using tensor representations in the hidden layers of a neural network along with simple inexpensive tensor operations leads to scalable equivariant networks. Further, these networks are universal approximators of functions equivariant to orthogonal groups. We find G-RepsNet to be competitive to EMLP on several tasks with group symmetries such as $O(5)$, $O(1,3)$, and $O(3)$ with scalars, vectors, and second-order tensors as data types. On image classification tasks, we find that G-RepsNet using second-order representations is competitive and often even outperforms sophisticated state-of-the-art equivariant models such as GCNNs (Cohen & Welling, 2016a) and $E(2)$-CNNs (Weiler & Cesa, 2019). To further illustrate the generality of our approach, we show that G-RepsNet is competitive to G-FNO (Helwig et al., 2023) and EGNN (Satorras et al., 2021) on N-body predictions and solving PDEs respectively, while being efficient. Code will be released at `https://github.com/merlresearch/G-RepsNets`

## 1 Introduction

Group equivariance plays a key role in the success of several popular architectures such as translation equivariance in Convolutional Neural Networks (CNNs) for image processing (LeCun et al., 1989), 3D rotational equivariance in Alphafold2 (Jumper et al., 2021), and equivariance to general discrete groups in Group Convolutional Neural Networks (GCNNs) (Cohen & Welling, 2016a).

But designing efficient equivariant networks can be challenging because they both require domain-specific knowledge and can be computationally inefficient. E.g., there are several works designing architectures for different groups such as the special Euclidean group $SE(3)$ (Fuchs et al., 2020), special Lorentz group $O(1,3)$ (Bogatskiy et al., 2020), discrete Euclidean groups (Cohen & Welling, 2016a; Ravanbakhsh et al., 2017), etc. Moreover, some of these networks can be computationally inefficient, prompting the design of simpler and lightweight equivariant networks such as EGNN (Satorras et al., 2021) for graphs and vector neurons (Deng et al., 2021) for point cloud processing.

---

[*]Sourya Basu was an intern at MERL when this work was performed

Finzi et al. (2021b) propose an algorithm to construct equivariant MLPs (EMLPs) for arbitrary matrix groups when the data is provided using tensor polynomial representations. This method directly computes the basis of the equivariant MLPs and requires minimal domain knowledge. However, using the computed equivariant basis can be computationally expensive, and it is impractical to use them for practical datasets such as images and point clouds, as noted by the authors (Finzi et al., 2021a). It is also noted in prior works (Fuchs et al., 2020; Thomas et al., 2018) that using equivariant basis, even for simple groups such as $SO(3)$, can be computationally expensive and it is impractical to use them to scale up to large datasets. Hence, we propose a lightweight construction of equivariant networks which is inexpensive, and yet is competitive to EMLPs for toy datasets and scales up to larger datasets of practical importance.

To this end, we introduce Group Representation Network (G-RepsNet), which replaces scalar representation from classical neural networks with tensor representations of different orders to obtain expressive equivariant networks. We use the same tensor polynomial representations as EMLP to represent the features in our network. But unlike EMLP, we only use inexpensive tensor operations such as tensor addition and tensor multiplication to construct our network. We show that even with these simple operations, we obtain a universal network for orthogonal groups. EMLPs are empirically known to be computationally expensive Further, we empirically show that G-RepsNet provides competitive results to existing state-of-the-art equivariant models and even outperforms them in several cases while having a simple and efficient design.

Our proposal generalizes Vector Neurons (Deng et al., 2021) which use first-order $O(3)$ tensor representations to obtain equivariance to the $O(3)$ group. In contrast, G-RepsNet is a construction which is equivariant to arbitrary matrix groups, universal for orthogonal groups, and uses higher-order tensor polynomial representations, while being computationally efficient. The main contributions as well as the summary of our results are detailed below.

1. We propose a novel lightweight construction of equivariant architectures. We call them G-RepsNets, which is are a class of computationally efficient architectures equivariant to arbitrary matrix groups and easy to construct.

2. We show that G-RepsNets are universal approximators of equivariant functions for orthogonal groups.

3. On synthetic datasets from Finzi et al. (2021b), we show that G-RepsNet is computationally much more efficient than EMLP and also performs competitively to EMLP across different groups such as $O(5)$, $O(3)$, and $O(1,3)$ using scalars, vectors, and second-order tensor representations.

4. We show that G-RepsNet with second-order tensor representations outperforms sophisticated state-of-the-art equivariant networks for image classification such as GCNNs (Cohen & Welling, 2016a) and $E(2)$-CNNs (Weiler & Cesa, 2019) when trained from scratch, and equitune (Basu et al., 2023b) when used with pretrained models.

5. G-RepsNet is competitive to G-FNO (Helwig et al., 2023) and EGNN (Satorras et al., 2021) on N-body predictions and solving PDEs, respectively, while being computationally efficient.

## 2  Related Work

**Parameter sharing** A popular method for constructing group equivariant architectures involves sharing learnable parameters in the network to guarantee equivariance, e.g. CNNs (LeCun et al., 1989), GCNNs (Cohen & Welling, 2016a; Kondor & Trivedi, 2018), Deepsets (Zaheer et al., 2017), etc. However, all these methods are restricted to discrete groups, unlike our work which can handle equivariance to arbitrary matrix groups.

**Steerable networks** Another popular approach for constructing group equivariant networks is by first computing a basis of the space of equivariant functions, then linearly combining these basis vectors to construct an equivariant network. This method can also handle continuous groups. Several popular architectures employ this method, e.g. steerable CNNs (Cohen & Welling, 2016b), $E(2)$-CNNs (Weiler & Cesa, 2019), Tensor Field Networks (Thomas et al., 2018), $SE(3)$-transformers (Fuchs et al., 2020), EMLPs (Finzi et al., 2021b) etc. But, these methods are computationally expensive and, thus, often replaced by efficient equivariant architectures for specific models, e.g., $E(n)$ equivariant graph neural networks (Satorras et al., 2021) for

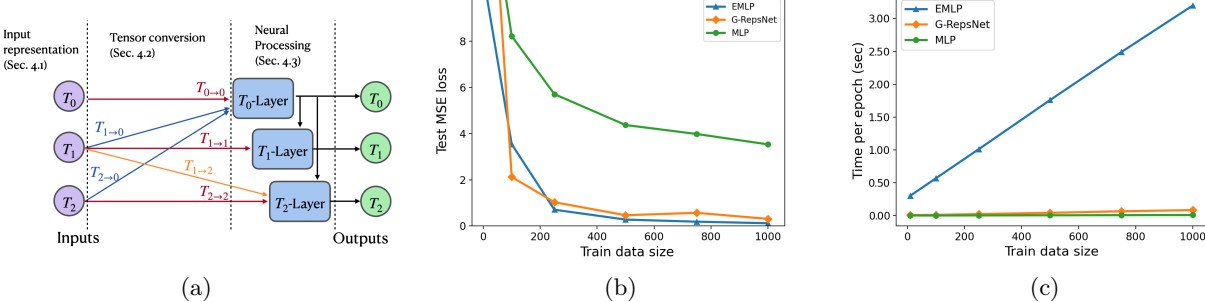

(a)  (b)  (c)

Figure 1: (a) Summary of G-RepsNet layer construction with example inputs of types $T_0, T_1$, and $T_2$, and outputs of the same types. Each layer consists of three subcomponents: i) input feature representation shown as $T_i$, ii) converting tensor types appropriately shown using arrows from $T_i$ to $T_j$, and iii) neural processing the converted tensors using appropriate neural networks, as discussed in Section 4. (b) and (c) provide comparisons of the loss and wall time of G-RepsNets with EMLPs (Finzi et al., 2021b) and MLPs for an O(3)-equivariant regression described in Section 5.1.

graphs and vector neurons (Deng et al., 2021) for point cloud processing. More comparisons with EMLPs are provided in Appendix A. Kondor et al. (2018) propose using steerable higher-order permutation representation to obtain a permutation-invariant graph neural networks. In contrast, we use higher-order tensors for arbitrary matrix groups, work with arbitrary base models such as CNNs, Fourier Neural Operators (FNOs) (Li et al., 2021), etc., and show that our architecture is a universal approximator for functions equivariant to orthogonal groups.

**Representation-based methods** A simple alternative to using steerable networks for continuous networks is to construct equivariant networks by simply representing the data using group representations, only using scalar weights to combine these representations, and using non-linearities that respect their equivariance. Works that use representation-based methods include vector neurons (Deng et al., 2021) for $O(3)$ group and universal scalars (Villar et al., 2021). Vector neurons are restricted to first-order tensors and Universal Scalars face scaling issues, hence, mostly restricted to synthetic experiments. More comparisons with universal scalars are provided in Appendix A.

**Frame averaging** Yet another approach to obtain group equivariance is to use frame-averaging (Yarotsky, 2022; Puny et al., 2021), where averaging over equivariant frames corresponding to each input is performed to obtain equivariant outputs. This method works for both discrete and continuous groups but requires the construction of these frames, either fixed by design as in Puny et al. (2021); Basu et al. (2023b) or learned using auxiliary equivariant neural networks as in Kaba et al. (2023). Our method is, in general, different from this approach since our method does not involve averaging over any frame or the use of auxiliary equivariant networks. For the special case of discrete groups, the notion of frame averaging is closely related to both parameter sharing as well as representation methods. Hence, in the context of equituning (Basu et al., 2023b), we show how higher-order tensor representations can directly be incorporated into their frame-averaging method.

## 3 Group and Representation Theory

A **group** is a set $G$ along with a binary operator '$\cdot$', such that the axioms of a group are satisfied: a) closure: $g_1 \cdot g_2 \in G$ for all $g_1, g_2 \in G$, b) associativity: $(g_1 \cdot g_2) \cdot g_3 = g_1 \cdot (g_2 \cdot g_3)$ for all $g_1, g_2, g_3 \in G$, c) identity: there exists $e \in G$ such that $e \cdot g = g \cdot e = g$ for any $g \in G$, d) inverse: for every $g \in G$ there exists $g^{-1} \in G$ such that $g \cdot g^{-1} = g^{-1} \cdot g = e$.

For a given set $\mathcal{X}$, a **group action** of a group $G$ on $\mathcal{X}$ is defined via a map $\alpha : G \times \mathcal{X} \mapsto \mathcal{X}$ such that $\alpha(e, x) = x$ for all $x \in \mathcal{X}$, and $\alpha(g_1, \alpha(g_2, x)) = \alpha(g_1 \cdot g_2, x)$ for all $g_1, g_2 \in G$ and $x \in \mathcal{X}$, where $e$ is the identity element of $G$. When clear from context, we write $\alpha(g, x)$ simply as $gx$. Given a function $f : \mathcal{X} \mapsto \mathcal{Y}$, we call the function $f$ to be $G$-**equivariant** if $f(gx) = gf(x)$ for all $g \in G$ and $x \in \mathcal{X}$.

Let $GL(m)$ represent the group of all invertible matrices of dimension $m$. Then, for a group $G$, a **linear group representation** of $G$ is defined as the map $\rho : G \mapsto GL(m)$ such that $\rho(g_1 g_2) = \rho(g_1)\rho(g_2)$ and $\rho(e) = I$, the identity matrix. A group representation of dimension $m$ is a linear group action on the vector space $\mathbb{R}^m$.

For a finite group $G$, the (left) **regular representation** $\rho$ over a vector space $V$ is a linear representation over $V$ that is freely generated by the elements of $G$, i.e., the elements of $G$ can be identified with a basis of $V$. Further, $\rho(g)$ can be determined by its action on the corresponding basis of $V$, $\rho(g) : h \mapsto gh$ for all $h \in G$. For designing G-RepsNet, note that the size of regular representation is equal to the size of $G$. The size of tensor product representations considered here, $m$, can be written as $m = |G| \times d$ for some integer $d$. We call the first dimension of size $|G|$ as the **group channel dimension**.

We call any linear group representation other than the regular representation as **non-regular group representation**. Examples of such representations include representations written as a Kronecker sum of irreducible representations (basis of a group representation). In the design of G-RepsNet, we use regular representation for finite groups and non-regular representations for continuous groups. Given some base linear group representation $\rho(g)$ for $g \in G$ on some vector space $V$, we construct **tensor representations** by applying Kronecker sum $\oplus$, Kronecker product $\otimes$, and tensor dual $^*$. Each of these tensor operations on the vector spaces leads to corresponding new group actions. The group action corresponding to $V^*$ becomes $\rho(g^{-1})^T$. Let $\rho_1(g)$ and $\rho_2(g)$ for $g \in G$ be group actions on vector spaces $V_1$ and $V_2$, respectively. Then, the group action on $V_1 \oplus V_2$ is given by $\rho_1(g) \oplus \rho_2(g)$ and that on $V_1 \otimes V_2$ is given by $\rho_1(g) \otimes \rho_2(g)$.

We denote the tensors corresponding to the base representation $\rho$ as $T_1$ tensors, i.e., tensors of order one, and $T_0$ denotes a scalar. In general, $T_m$ denotes a tensor of order $m$. Further, Kronecker product of tensors $T_m$ and $T_n$ gives a tensor $T_{m+n}$ of order $m + n$. We use the notation $T_m^{\otimes r}$ to denote $r$ times Kronecker product of $T_m$ tensors. Kronecker sum of two tensors of types $T_m$ and $T_n$ gives a tensor of type $T_m \oplus T_n$. Finally, Kronecker sum of $r$ tensors of the same type $T_m$ is written as $rT_m$.

$O(n)$ refers to the orthogonal group which is the group of dimension $n$ that preserves the distance in Euclidean space of dimension $n$. The group elements of $O(n)$ can be identified with orthogonal matrices $Q$ of dimension $n \times n$ which satisfy $Q^T Q = QQ^T = I_n$, where $I_n$ is the $n \times n$ identity matrix. Similarly, the Lorentz group $O(1, n)$ is the group of all isometries of $n$-dimensional spacetime that leave the origin fixed, where the distance is computed using the Minkowski metric.

## 4 G-RepsNet Architecture

Here, we describe the general design of the G-RepsNet architecture. Each layer of G-RepsNet consists of three subcomponents: i) representing features using appropriate tensor representation (Section 4.1), ii) converting tensor types of the input representation (Section 4.2), and iii) processing these converted tensors (Section 4.3). Finally, Section 4.4 discusses some properties of our network along with existing architectures that are special cases of G-RepsNet. Now we describe these subcomponents in detail.

### 4.1 Input Feature Representations

We employ two techniques to obtain input tensor representations for networks with regular and non-regular representations as described below.

**Regular representation:** Regular representation is favorable to use for small finite groups, e.g., cyclic group $C_n$ of discrete rotations of $\frac{360}{n}$ degrees. For input features for regular representations, we simply use the input features obtained from the $E(2)$-CNNs (Weiler & Cesa, 2019), but any regular representation works with our model. Thus, if we are given an image of dimension $B \times C \times H \times W$, the $T_1$ regular representation is of dimension $|G| \times B \times C \times H \times W$, where $|G|, B, C, H, W$ are the group channel dimension, batch size, channel size, height, and width, respectively. Similarly, for any tensor of type $T_i$, the group channel dimension of size $|G|^i$.

**Non-regular representation:** Non-regular representation is useful for all continuous groups as well as large finite groups, e.g. $SO(n)$ group of rotations, $S(n)$ group of permutations. For non-regular representations,

usually, the data is naturally provided in suitable tensor representations, e.g., position and velocity data of particles from the synthetic datasets in Finzi et al. (2021b) are provided in the form of Kronecker sum of irreducible representations of groups such as $O(n)$. In all our experiments using continuous groups, the inputs are already provided as tensor representations using the appropriate irreducible representations. Here, we call the dimension of the matrix representation of the group as the group channel dimension. E.g., for the $SO(2)$ group elements represented as $2 \times 2$ matrices, we have the group channel dimension equals to 2.

## 4.2 Tensor Conversion

Crucial to our architecture is the tensor conversion component. The input to each layer in Fig. 1a is given as a concatenation of tensors of varying orders. But $T_i$-layer in Fig. 1a only processes tensors of order $T_i$. Thus, to process tensors of order $T_j$, $j \neq i$, they must first be converted to tensors of order $T_i$ and then passed to the $T_i$-layer. Our tensor conversion algorithm is described next.

When $i > j > 0$, we convert tensors of type $T_j$ to tensors of type $T_i$ by first writing $i = kj + r$, where $k = \lfloor i/j \rfloor$. Then, we obtain $T_i$ from $T_j$ and $T_1$ as $T_j^{\otimes k} \otimes T_1^{\otimes r}$. When using non-regular representations, we assume that the input to the G-RepsNet model always consists of some tensors with $T_1$ representations, which is not a strong assumption that helps keep our construction simple and also encompasses all experiments from Finzi et al. (2021b). We do not convert tensors of type $T_j$ to $T_i$ for $0 < i < j$ as (a) it requires tensor decomposition, which can be expensive in practice, and (b) we already obtain universality for orthogonal groups without it.

When $i = 0$, we convert each input of type $T_j$ to type $T_0$ by using an appropriate invariant operator, e.g. Euclidean norm for Euclidean groups, or averaging over the group channel dimension for regular groups. These design choices keep our design lightweight as well as expressive as we show both theoretically as well as empirically. Details on processing these inputs are described next.

## 4.3 Neural Processing

Now we discuss how the various $T_i$-layers are constructed and how they process the input tensor features that have been converted to $T_i$ tensor types. We use different techniques for regular and non-regular tensor representations.

**Regular representation:** Recall that regular representations for tensors of type $T_i$ have group channel dimensions equal to $|G|^i$, where $|G|$ is the size of the group. For tensors of dimension $(|G|^i \times B) \times C \times H \times W$, we treat the group channel dimension just like the batch dimension and process the $(|G|^i \times B)$ inputs in parallel through the same model. Here we are free to choose any model of our choice for any of the $T_i$-layers, e.g., MLP, CNNs, FNOs, etc. We call these models of choice our **base model** just like used in frame-averaging (Puny et al., 2021) and equitune (Basu et al., 2023b).

**Non-regular representation:** Here, we impose certain restrictions on what models can be used for $T_i$-layers and how to use them. First, the $T_0$-layer passes all the tensors of type $T_0$ or scalars through a neural network such as an MLP or a CNN. Since the inputs are invariant scalars, the outputs are always invariant and thus, there are no restrictions on the neural network used for the $T_0$-layer, i.e., they may also use non-linearities. Now we describe how to process the tensors of type $T_i$ for $i > 0$.

Let us call the output from the $T_0$-layer as $Y_{T_0}$. For a $T_i$-layer with $i > 0$, we first multiply the input with a learnable weight matrix along the *data dimension*, i.e. the dimensions other than the group channel dimension and batch dimension, with no point-wise non-linearities or bias terms. This ensures that the output is equivariant just as in Vector Neurons (Deng et al., 2021). E.g., if the input to a $T_i$-layer for the $SO(2)$ group is $B \times 2 \times n$, then we multiply the input with a matrix of dimension $n \times m$ to get an output of dimension $B \times 2 \times m$. Let us call the output from this linear layer as $H_{T_i}$.

Then, to mix the $T_i$ tensors with the $T_0$ tensors better, we update $H_{T_i}$ as $H_{T_i} = H_{T_i} * \frac{Y_{T_0}}{\text{inv}(H_{T_i})}$, where $\text{inv}(\cdot)$ is a group-invariant function such as the Euclidean norm for a Euclidean group. We note here that this mixing function bears similarity to the bilinear layers used in EMLPs (Finzi et al., 2021b). However, we

differ in its motivation. In EMLPs, the motivation is solely to enhance performance, whereas, in our design, it is motivated from the to make the design universal for orthogonal groups (see Appendix C.

Finally, we pass $H_{T_i}$ through another linear layer without any bias or pointwise non-linearities to obtain $Y_{T_i}$. This *mixing* of various tensor types is crucial to make our network expressive and is required for the universality of our network.

Both (a) tensor conversion, that allows higher order tensor features and (b) tensor mixing that combines information among tensors of different orders/types are novel aspects of the proposed G-RepsNet architecture that generalize earlier works (see Appendix A) (Deng et al., 2021; Zaheer et al., 2017; Basu et al., 2023b). Our proofs on universality of G-RepsNets to orthogonal groups (see Appendix C) make use of this tensor-mixing step to show that no theoretical representation ability is lost in spite of the simple construction, in the case of orthogonal groups.

### 4.4 Properties

**Equivariance:** For regular representations, any group action applied to the input appears as a permutation in the group channel dimension. Further, since the data is processed in parallel along the group channel dimension, the output permutes accordingly, making our model equivariant. For non-regular representation, the $T_0$ layer only processes invariant tensors and, hence, preserves equivariance of the overall model. Moreover, the $T_i$-layers simply perform a linear combination of tensors, making the overall model equivariant. A proof for equivariance of our model is given in Appendix B.

**Universality:** Here, we show that our models are universal approximators of equivariant functions for orthogonal groups. This ensures that our models are expressive. For models constructed for regular representations, it is easy to verify that there exist G-RepsNets that are universal approximators of equivariant functions. To that end, note that restricting G-RepsNet to only $T_1$ tensors and taking an average with group inverses along the group channel dimension gives group symmetrization in equitune (Basu et al., 2023b; Yarotsky, 2022). It is well known that the symmetrization of universal approximators such as MLPs give universal approximators of equivariant functions (Yarotsky, 2022). It follows that G-RepsNets are universal approximators of equivariant functions for regular representations. Note that even though our models using features of type $T_1$ themselves are universal approximators, we illustrate empirically that higher order tensors significantly boost the performance of G-RepsNet with regular representation.

For non-regular representations, we provide simple constructive proofs showing the universality properties of the G-RepsNet architecture. We first show that G-RepsNet can approximate arbitrary invariant scalar functions of vectors from $O(d)$ and $O(1,d)$ groups. Then, we extend the proof to vector-valued functions for the same groups. First, recall the Fundamental Theorem of Invariant Theory for $O(d)$ as described in Lemma .1.

**Lemma 1** (Weyl (1946)). *A function of vector inputs returns an invariant scalar if and only if it can be written as a function only of the invariant scalar products of the input vectors. That is, given input vectors $(X_1, X_2, \ldots, X_n)$, $X_i \in \mathbb{R}^d$, any invariant scalar function $h : \mathbb{R}^{d \times n} \mapsto \mathbb{R}$ can be written as*

$$h(X_1, X_2, \ldots, X_n) = f(\langle X_i, X_j \rangle_{i,j=1}^n), \tag{1}$$

*where $\langle X_i, X_j \rangle$ denotes the inner product between $X_i$ and $X_j$, and $f$ is an arbitrary function.*

As mentioned in Villar et al. (2021), a similar result holds for the $O(1,d)$ group. In Thm. 1, we show that G-RepsNet can approximate arbitrary invariant scalar functions for $O(d)$ or $O(1,d)$ groups. The main idea of the proof is to show that G-RepsNet can automatically compute the necessary inner products $\langle X_i, X_j \rangle$ in equation 1 and the function $f$ in equation 1 can be approximated using an MLP in the $T_0$ layer. Detailed proof is provided in Appendix C.

**Theorem 1.** *For given $T_1$ inputs $(X_1, X_2, \ldots, X_n)$ corresponding to $O(d)$ or $O(1,d)$ group, $X_i \in R^d$, any invariant scalar function $h : \mathbb{R}^{d \times n} \mapsto \mathbb{R}$, there exists a G-RepsNet model that can approximate $h$.*

Similarly, this result can be extended to vector functions as described in Thm. 2. The proof for Thm. 2 is also constructive and is provided in Appendix C.

**Theorem 2.** *For given $T_1$ inputs $(X_1, X_2, \ldots, X_n)$ corresponding to $O(d)$ or $O(1, d)$ group, $X_i \in R^d$, any equivariant vector function $h : \mathbb{R}^{d \times n} \mapsto \mathbb{R}^d$, there exists a G-RepsNet model that can approximate $h$.*

Finally, in Appendix A.2 we show how several popular models such as Vector Neurons (Deng et al., 2021), harmonic networks (Worrall et al., 2017), Deepsets (Zaheer et al., 2017), and equitune (Basu et al., 2023b) are special cases of G-RepsNet.

## 5 Applications and Experiments

We conduct several experiments to show that our model is competitive with state-of-the-art equivariant models across various domains while also being easy to design and computationally efficient.

For non-regular representation, we provide two experiments: i) in Section 5.1 we compare G-RepsNet with EMLP on synthetic datasets, which encompass equivariance to several groups such as $O(5)$, $O(3)$, and $O(1, 3)$ and involves tensors of different orders; ii) in Section 5.2 we compare G-RepsNet with EGNN on N-body dynamics prediction. Note that the groups considered are restricted to orthogonal groups, even though G-RepsNets work for arbitrary matrix groups just like EMLPs. This is because we directly take the datasets from Finzi et al. (2021b); Satorras et al. (2021), which are restricted to orthogonal groups.

For regular representation, we provide two more experiments: i) for image classification we show in Section 5.3 that G-RepsNet with higher-order tensors and CNNs as the base model can outperform popular equivariant models such as GCNNs and $E(2)$-CNNs when trained from scratch and equitune when used with pretrained models; ii) for solving PDEs, in Appendix 5.4 we construct a G-RepsNet with FNOs as the base model, where we find G-RepsNets is competitive to more sophisticated equivariant models such as G-FNOs (Helwig et al., 2023) while being much faster than them.

### 5.1 Comparison with EMLPs

**Datasets:** We consider three regression tasks from Finzi et al. (2021b): $O(5)$-invariant task, $O(3)$-equivariant task, and $O(1, 3)$ invariant task. In $O(5)$-invariant regression, we have input $X = \{x_i\}_{i=1}^2$ of type $2T_1$ and output $f(x_1, x_2) = \sin(\|x_1\|) - \|x_2\|^3 / 2 + \frac{x_1^T x_2}{\|x_1\|\|x_2\|}$ of type $T_0$. Then, for $O(3)$-equivariant task we have input $X = \{(m_i, x_i)\}_{i=1}^5$ of type $5T_0 + 5T_1$ corresponding to 5 masses and their positions. The output is the inertia matrix $\mathcal{I} = \sum_i m_i (x_i^T x_i I - x_i x_i^T)$ of type $T_2$. Finally, for the $O(1, 3)$-equivariant task, we use the electron-muon scattering $(e^- + \mu^- \rightarrow e^- + \mu^-)$ task from Finzi et al. (2021b), originally from Bogatskiy et al. (2020). Here, the input is of type $4T_{(1,0)}$ corresponding to the four momenta of input and output electron and muon, and the output is the matrix element of type $T_{(0,0)}$ (Finzi et al., 2021b).

**Model design:** For all the experiments here, we design G-RepsNet using different tensor representations in each of the models depending on the application. We call the number of tensors in a hidden layer as its channel size. We fix a channel size of 100.

$O(5)$**-invariant model:** The input consists of two tensors of $T_1$ type that are passed through the first layer consisting of $T_0$-layers and $T_1$-layers similar to vector neurons shown in Fig. 6a, but our design differs from vector neurons in that we use simple Euclidean norm to compute the $T_0$ converted tensors instead of dot product used by vector neurons. All $T_i$ layers are made of MLPs. The number of output tensors is equal to the channel size, and the channel sizes used for our experiments in discussed in Section 5.1. This is followed by three similar layers consisting of $T_0$-layers and $T_1$-layers, all of which take as input $T_1$ tensors, and output tensors of the same type. Additionally, these layers use residual connections as shown in Fig. 5a. Finally, the $T_1$ tensors are converted to $T_0$ tensors by taking their norms, which are passed through a final $T_0$-layer that gives the output.

More precisely, in our experiments, we consider a model with 5 learnable linear layers with no bias terms, where the dimensions of the layers are $(2 \times 100, 100 \times 100, 100 \times 100, 100 \times 100, 100 \times 1)$. The input of type $2T_1$ is of dimension $(2 \times 5)$. The input is first passed through the first layer of dimension $2 \times 100$ to obtain a hidden layer output of type $100T_1$. Then, this output is also converted to type $100T_0$ by simply taking the norm. Thus, we have a tensor of type $100T_0 + 100T_1$. Finally, we convert this tensor of type $100(T_0 + T_1)$ to $100T_1$ by simply multiplying the $100T_0$ scalars with the $100T_1$ vectors. This is basically a simplified version

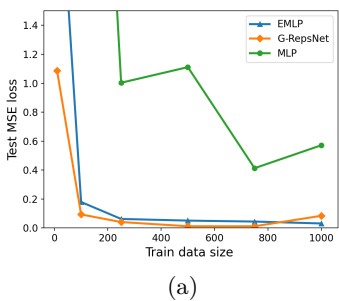 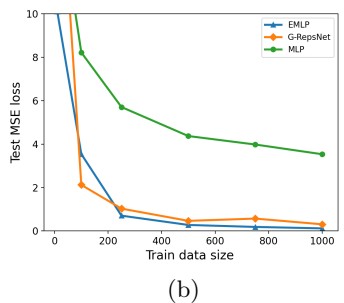 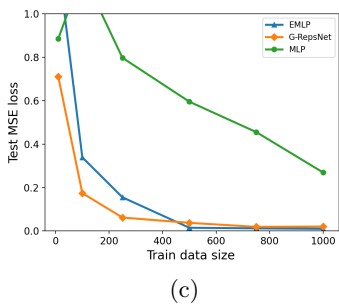

(a)                                   (b)                                   (c)

Figure 2: Comparison of G-RepsNets with EMLPs (Finzi et al., 2021b) and MLPs for (a) $O(5)$-invariant synthetic regression task with input type $2T_1$ and output type $T_0$, (b) $O(3)$-equivariant regression with input as masses and positions of 5 point masses using representation of type $5T_0 + 5T_1$ and output as the inertia matrix of type $T_2$, (c) $SO(1,3)$-invariant regression computing the matrix element in electron-muon particle scattering with input of type $4T_1$ and output of type $T_0$.

Table 1: Comparison of MLP, G-RepsNet and EMLP on tasks using non-regular representation. Train time per epoch (in seconds), test time per sample (in milliseconds) and number of parameters for models with the same channel size of 384 for training datasets of size 1000 are provided.

| Model
Task | MLP | | | G-RepsNet | | | EMLP | | |
|---|---|---|---|---|---|---|---|---|---|
| | # params | Train time (s) | Test time (ms) | # params | Train Time (s) | Test time (ms) | # params | Train time (s) | Test time (ms) |
| $O(5)$-invariant | 300,289 | 0.0083 | 0.33 | 30,300 | 0.013 | 0.52 | 480,694 | 3.00 | 7.73 |
| $O(3)$-equivariant | 307,209 | 0.0087 | 0.32 | 233,200 | 0.084 | 1.55 | 572,121 | 3.19 | 7.13 |
| $SO(1,3)$-invariant | 302,593 | 0.0080 | 0.33 | 30,500 | 0.049 | 1.05 | 446,688 | 2.86 | 8.01 |

of the tensor mixing process described in Section 4.3. This gives a tensor of type $100T_1$, which is the input for the next layer. We repeat the same process of converting to $T_0$ and back to $T1$ for the next two layers. For the final two layers, we convert all the tensors to scalars of type $100T_0$ and process through the last two layers and use ReLU activation function in between.

$O(3)$-**equivariant model:** The model contains four layers that take in 5 input tensors each of type $T_0 + T_1$ and output a single tensor of type $T_2$. A detailed description of the four layers are as follows.

*First layer:* Let the input and output of the first layer be $X_{T_0}, X_{T_1}$ and $H_{T_0}, H_{T_1}, H_{T_2}$, respectively. Here, $X_{T_i}$ denotes tensors of type $T_i$ and similarly for $H_{T_i}$.

To compute $H_{T_0}$, we first convert $X_{T_1}$ to type $T_0$ by taking its norm and concatenating it with $X_{T_0}$. Let us assign this concatenated value to $H_{T_0}$. Then, the final value of $H_{T_0}$ is obtained by passing $H_{T_0}$ through two linear layers with a ReLU activation in between.

To compute $H_{T_1}$, we simply perform $W_2(H_{T_0} * W_1(X_{T_1})/\|W_1(X_{T_1})\|)$ as the tensor mixing process from Section 4.3, where $W_1, W_2$ are single linear layers with no bias terms. To compute $H_{T_2}$, we first convert $X_{T_0}$ to type $T_2$ by multiplying it with an identity matrix of dimension of $X_{T_2}$. Let us call this $H_{T_{20}}$. Then, we convert $X_{T_1}$ to type $T_2$ by taking the outer product with itself. Let us call this $H_{T_{21}}$. We concatenate $H_{T_{20}}$ and $H_{T_{21}}$, and call this $H_{T_2}$. Then, we update $H_{T_2}$ as follows. We simply perform $W_2(H_{T_0} * W_1(H_{T_2})/\|W_1(H_{T_2})\|)$ as the tensor mixing process from Section 4.3, where $W_1, W_2$ are single-layered linear layers with no bias terms. In all cases, the number of tensors obtained is equal to the channel size used for the experiments discussed in Section 5.1.

*Second and third layers:* The first layer above is followed by two layers of input and output types $T_0 + T_1 + T_2$. Let the input and output of the this type of layer be $X_{T_0}, X_{T_1}, X_{T_2}$ and $H_{T_0}, H_{T_1}, H_{T_2}$, respectively. Here, $X_{T_i}$ denotes tensors of type $T_i$ and similarly for $H_{T_i}$. To compute $H_{T_0}$, we first convert $X_{T_1}$ and $X_{T_2}$ to type $T_0$ by taking its norm and concatenate it to $X_{T_0}$. Let us assign this concatenated value to $H_{T_0}$. Then, the final value of $H_{T_0}$ is obtained by passing $H_{T_0}$ through two linear layers with a ReLU activation in between.

The rest of the computations for obtaining $H_{T_1}$ and $H_{T_2}$ are identical to the first layer, which is described below for completeness.

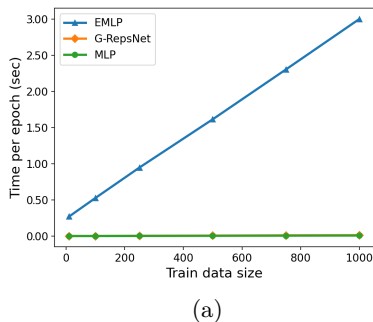 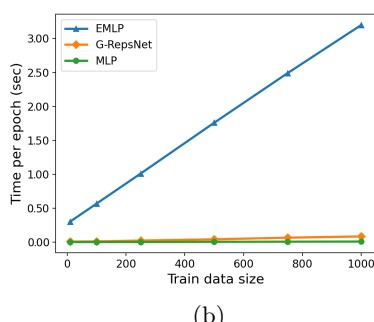 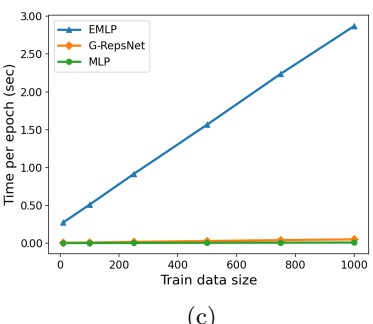

| (a) | (b) | (c) |

Figure 3: Times per epoch (in seconds) for different MLPs, G-RepsNets, and EMLPs for varying dataset sizes. Here the figures correspond to the experiments described next. a) $O(5)$-invariant synthetic regression task with input type $2T_1$ and output type $T_0$, (b) $O(3)$-equivariant regression with input as masses and positions of 5 point masses using representation of type $5T_0 + 5T_1$ and output as the inertia matrix of type $T_2$, (c) $SO(1,3)$-invariant regression computing the matrix element in electron-muon particle scattering with input of type $4T_1$ and output of type $T_0$.

Table 2: Comparison of EGNN and G-RepsGNN on the $O(3)$-equivariant 5-body dynamics prediction task. Note that G-RepsGNN is constructed by simply replacing the representation in the GNN architecture from Gilmer et al. (2017) with $T_1$ representations along with tensor mixing from Section 4.3, whereas EGNN is a specialized GNN designed for $E(n)$-equivariant tasks.

| Model | Test Loss | Forward Time (ms) | Number of parameters |
|---|---|---|---|
| EGNN | 0.0069 | 1.76 | 69215 |
| G-RepsGNN (ours) | 0.0049 | 2.02 | 106288 |

To compute $H_{T_1}$, we simply perform $W_2(H_{T_0} * W_1(X_{T_1})/\|W_1(X_{T_1})\|)$ as the mixing process from Section 4.3, where $W_1, W_2$ are single linear layers with no bias terms.

To compute $H_{T_2}$, we first convert $X_{T_0}$ to type $T_2$ by multiplying it with an identity matrix of dimension of $X_{T_2}$. Let us call this $H_{T_{2_0}}$. Then, we convert $X_{T_1}$ to type $T_2$ by taking the outer product with itself. Let us call this $H_{T_{2_1}}$. We concatenate $H_{T_{2_0}}$, $H_{T_{2_1}}$, and $X_{T_2}$, and call this $H_{T_2}$. Then, we update $H_{T_2}$ as follows. We simply perform $W_2(H_{T_0} * W_1(H_{T_2})/\|W_1(H_{T_2})\|)$ as the tensor mixing process from Section 4.3, where $W_1, W_2$ are single-layered linear layers with no bias terms.

These layers also use residual connections similar to the ones shown in Fig. 5a.

*Fourth layer:* Finally, the $T_2$ tensors of the output of the penultimate layer are passed through a final $T_2$ layer, which gives the final output.

$O(1,3)$**-invariant model** This design is identical to the design of the $O(5)$-invariant network above except for a few changes: a) the invariant tensors is obtained using Minkowski norm instead of the Euclidean norm, b) the number of channels are decided by the number of channels chosen for this specific experiment in Section 5.1.

**Experimental results:** We train MLPs, EMLPs, and G-RepsNet on the datasets discussed above for 100 epochs. Further details on the hyperparameters are given in Section D.1. From Fig. 2, we find that across all the tasks, G-RepsNets perform competitively to EMLPs and significantly outperform non-equivariant MLPs. Moreover, Fig. 3 and Tab. 1 show that G-RepsNets are computationally much more efficient than EMLPs, while being only slightly more expensive than naive MLPs. This shows that G-RepsNet can provide competitive performance to EMLPs on equivariant tasks. Moreover, the lightweight design of G-RepsNets motivates its use in larger datasets.

Table 3: Table shows mean (std) of classification accuracies on Rot90-CIFAR10 dataset for $T_1$-G-RepsCNN, $T_2$-G-RepsCNN, GCNN, and $T_2$-G-RepsGCNN for 100 epochs. The base model used here is Resnet18, and the $T_2$ layers in the $T_2$ models are added only in the last layer as described in Appendix D.2. Results are over 3 seeds.

| Dataset \ Model | CNN | $T_1$-G-RepsCNN | $T_2$-G-RepsCNN | GCNN | $T_2$-G-RepsGCNN |
|---|---|---|---|---|---|
| Rot90-CIFAR10 | 72.8 (0.2) | 79.5 (0.2) | **80.4 (0.4)** | 73.8 (0.9) | **76.6 (0.5)** |

### 5.2 Modelling a Dynamic N-Body System with GNNs

**Dataset details:** We consider the problem of predicting the dynamics of $N$ charged particles given their charges and initial positions, where the symmetry group for equivariance is the orthogonal group $O(3)$. Each particle is placed at a node of a graph $\mathcal{G} = \{\mathcal{V}, \mathcal{E}\}$, where $\mathcal{V}$ and $\mathcal{E}$ are the sets of vertices and edges. We use the N-body dynamics dataset from Satorras et al. (2021), where the task is to predict the positions of $N = 5$ charged particles after $T = 1000$ steps given their initial positions $\in \mathbb{R}^{3 \times 5}$, velocities $\in \mathbb{R}^{3 \times 5}$, and charges $\in \{-1, 1\}^5$.

**Model design and experimental setup:** Let the edge attributes of $\mathcal{G}$ be $a_{ij}$, and let $h_i^l$ be the node feature of node $v_i \in \mathcal{V}$ at layer $l$ of a message passing neural network (MPNN). An MPNN as defined by Gilmer et al. (2017) has an edge update, $m_{ij} = \phi_e(h_i^l, h_j^l, a_{ij})$ and a node update $h_i^{l+1} = \phi_h(h_i^l, m_i)$, $m_i = \sum_{j \in \mathcal{N}(i)} m_{ij}$, where $\phi_e$ and $\phi_h$ are MLPs corresponding to edge and node updates, respectively.

We design G-RepsGNN by making small modifications to the MPNN architecture. In our model, we use two edge updates for $T_0$ and $T_1$ tensors, respectively, and one node update for $T_1$ update. The two edge updates are $m_{ij,T_0} = \phi_{e,T_0}(\|h_i^l\|, \|h_j^l\|, a_{ij})$, $m_{ij,T_1} = \phi_{e,T_1}(h_i^l, h_j^l, a_{ij})$, where $\|\cdot\|$ obtains $T_0$ tensors from $T_1$ tensors for the Euclidean group, $\phi_{e,T_0}(\cdot)$ is $T_0$-layer MLP, and $\phi_{e,T_1}(\cdot)$ is a $T_1$-layer made of an MLP without any pointwise non-linearities or biases. The final edge update is obtained as $m_{ij} = m_{ij,T_1} * m_{ij,T_0} / \|m_{ij,T_1}\|$. Finally, the node update is given by $h_i^{l+1} = \phi_{h,T_1}(h_i^l, m_i)$, where $m_i = \sum_{j \in \mathcal{N}(i)} m_{ij}$ and $\phi_{h,T_1}(\cdot)$ is an MLP without any pointwise non-linearities or biases. Thus, the final node update is a $T_1$ tensor. We compare G-RepsGNN with EGNN (Satorras et al., 2021), which is a popular equivariant graph neural network.

We closely followed Satorras et al. (2021) to generate the dataset: we used 3000 trajectories for train, 2000 trajectories for validation, and 2000 for test. Both EGNN and G-RepsGNN models have 4 layers and were trained for 10000 epochs, same as in Satorras et al. (2021).

**Results and Observations:** From Tab. 2, we find that even though EGNN is a specialized architecture for the task, G-RepsGNN performs competitively to EGNN. Note that here the comparison is made to EGNN since it is a computationally efficient expressive equivariant model just like G-RepsGNN, although restricted for processing graphs. Here our goal is not to achieve state-of-the-art results on this task but to simply show that our model is competitive with popular models even with minimal design changes to the non-equivariant base model MPNN. Further results of test losses and forward times for various other models are reported in Tab. 7 in Appendix E.1 for completeness. Since G-RepsGNN has a comparable computational complexity to EGNN, it is computationally much more efficient than many specialized group equivariant architectures that use spherical harmonics for $E(n)$-equivariance as noted from Tab. 7.

### 5.3 Second-Order Image Classification

We perform two sets of experiments: i) in the first, we train various image classification models from scratch and compare them with G-RepsNet and ii) we perform equivariant finetuning of non-equivariant models with higher-order tensors, hence extending the equivariant finetuning method of Basu et al. (2023b) to second-order tensors.

**Dataset:** For ablation studies to understand the effect of second-order tensors in image classification and for experiments involving training from scratch, we test on different datasets obtained by applying random rotations to the CIFAR10 dataset. When the random rotations are a multiple of $\frac{360}{n}$ for integer $n$, we call the dataset Rot$\frac{360}{n}$-CIFAR10, else if the rotations are by arbitrary angles in $(0, 360]$, we simply call it Rot-CIFAR10. These rotations are applied to ensure that the dataset exhibits the $C_n$ of multiples of

Table 4: Table shows mean (std) test accuracies for equituning using a pretrained Resnet with Rot90-CIFAR10 and Galaxy10. For the $T_2$ models, the $T_2$ layers are added only in the last layer.

| Dataset\Model | Finetune | $T_1$-Equitune | $T_2$-Equitune |
|---|---|---|---|
| Rot90-CIFAR10 | 82.7 (0.5) | 88.1 (0.3) | **89.6 (0.3)** |
| Galaxy10 | 76.9 (3.2) | 79.3 (1.6) | **80.7 (4.0)** |

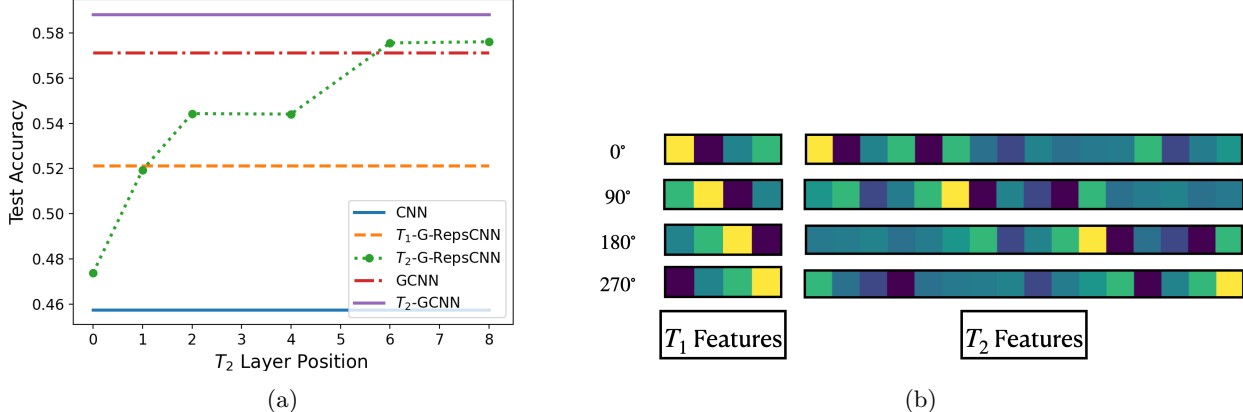

(a)            (b)

Figure 4: In (a), we analyze the performance of a rot90-equivariant CNN with 3 convolutional layers and 5 fully connected layers on rot90-CIFAR10. Here, $T_2$ representations are introduced in layer $i \in [1, \ldots, 8]$. We find that using $T_2$ representations in the final layers of the CNN easily outperforms non-equivariant CNNs as well as traditional equivariant representations with $T_1$ representations. (b) shows the $T_1$ and $T_2$ features obtained from one channel of a pretrained Resnet corresponding to $T_1$-equitune and $T_2$-equitune, respectively.

$\frac{360}{n}$-degree rotations or $SO(2)$ symmetry, which helps test the working of equivariant networks. For our experiments on equivariant finetuning, we test on Rot90-CIFAR10 and Galaxy10 without any rotations. We do not apply rotations on Galaxy10 since it naturally has the $C_4$ symmetry.

**Experimental setup and model design:** We first design a rot90-equivariant CNN with 3 conv layers followed by 5 fully connected layers and train it from scratch on CIFAR10 with random rotations. We use $T_1$ representations for the first $i$ layers and use $T_2$ representations for the rest, where the $T_2$ representations are obtained by simple outer product of the $T_1$ representation in the group channel dimension. Fig. 5 shows a simple way to add residual connections in G-RepsNet as well as a general architecture for $T2$-G-RepsCNN. It is easy to verify the equivariance is maintained for both $T_1$ and $T_2$ for regular representations. It is also easy to see that because of our simple design, the number of parameters for the base CNN, $T_1$-G-RepsCNN and $T_2$-G-RepsCNN are exactly the same (about 150K parameters, and takes only about 0.8ms for a forward pass on an Nvidia A5500). We train each model for 10 epochs. The results reported in Fig. 4a indicate that using $T_2$ representations in the later layers of the same network significantly outperforms both non-equivariant as well as equivariant $T_1$-based CNNs. Fig. 4b provides a visualization of the $T_2$ features compared to the $T_2$ features. The $T_1$ or $T_2$ features are convered to $C_n$-invariant $T_0$ features by taking the mean along the group dimension. Hyperparameter values are provided in Appendixs D.2. We compare with baseline equivariant architectures such as GCNNs (Cohen & Welling, 2016a) and $E(2)$-CNNs (Weiler & Cesa, 2019).

First, for comparison with GCNNs, we use CNN as well as GCNN as our base model for constructing G-RepsNets. We use the resnet architecture (He et al., 2016) as our CNNs. For GCNNs and E(2)-CNNs, we simply replace the convolutions with group convolutions (Cohen & Welling, 2016a) and E(2)-CNNs (Weiler & Cesa, 2019), respectively in the same CNN and adjust the channel sizes to ensure a nearly equal number of parameters.

We design two G-RepsCNN architectures: a) $T_1$-G-RepsCNN, where each layer has a $T_1$ representation, and b) $T_2$-G-RepsCNN, where all the layers except the last layer use $T_1$ representation and the last layer uses $T_2$ representation. Using GCNN as the base model, we construct $T_2$-G-RepsGCNN, by simply replacing the CNN model with a GCNN in the $T_2$-G-RepsCNN. Note that we do not construct $T_1$-GCNN as it results in

Table 5: Table shows mean (std) of classification accuracies on Rot-CIFAR10 dataset (CIFAR10 with random rotations in (-180°, +180°]) for various G-RepsCNNs and $E(2)$-CNNs with different group equivariances, and tensor orders. he base model used here is Resnet18, and the T2 layers in the T2 models are added only in the last layer as described in Appendix D.2. All models are trained for 100 epochs and results are over 3 fixed seeds.

| Model | Equivariance | Tensor Orders | Test Acc. |
|---|---|---|---|
| CNN | – | – | 65.21 (0.4) |
| $T_1$-G-RepsCNN | C8 | $(T_1)$ | 73.4 (0.4) |
| $T_2$-G-RepsCNN | C8 | $(T_1, T_2)$ | **73.8 (0.4)** |
| $E(2)$-CNN | C8 | $(T_1)$ | 49.6 (1.6) |
| $T_2$-G-Reps$E(2)$-CNN | C8 | $(T_1, T_2)$ | 57.3 (1.4) |
| $T_1$-G-RepsCNN | C16 | $(T_1)$ | 73.8 (0.1) |
| $T_2$-G-RepsCNN | C16 | $(T_1, T_2)$ | **75.2 (0.5)** |
| $E(2)$-CNN | C16 | $(T_1)$ | 46.8 (0.8) |
| $T_2$-G-Reps$E(2)$-CNN | C16 | $(T_1, T_2)$ | 55.4 (1.9) |

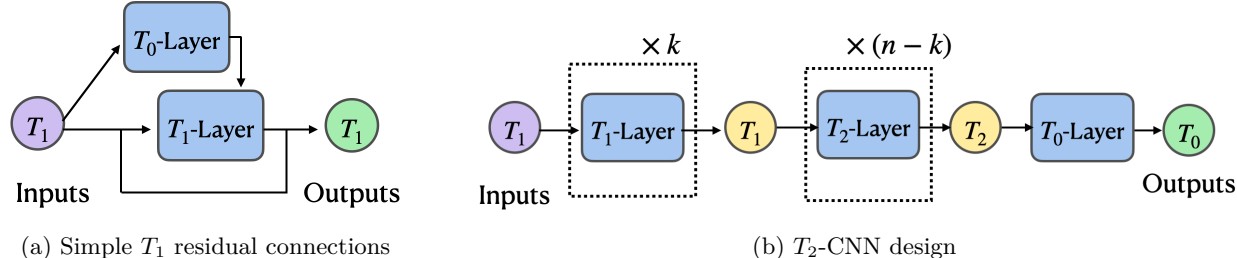

(a) Simple $T_1$ residual connections       (b) $T_2$-CNN design

Figure 5: (a) shows a simple way to add residual connections in G-RepsNet. (b) shows the architecture used for $T_2$ CNNs and equituning, where the first $k$ layers are made of $T_1$-layers to extract features, then the extracted features are converted in $T_2$ tensors, which are then processed by $T_2$-layers. Finally $T_0$ tensors, i.e., scalars are obtained as the final output.

the same model as GCNN. These models are compared on the Rot90-CIFAR10 dataset. The base architecture for constructing the G-RepsCNNs is ResNet18 and the $T_2$ layers are only used in the last layer. Here, we note that, if we decompose the features in G-RepsCNNs in terms of irreps, our design can be seen as a special case of Steerable CNNs (Weiler & Cesa, 2019), except that for constructing G-RepsCNN, we do not need to analytically solve for the equivariant layers. Moreover, in some cases such as SE(3)-tranformers Fuchs et al. (2020), even when the analytical solution is given, they can be computationally expensive.

For comparison with $E(2)$-CNNs, we perform a similar comparison as that with GCNNs. Here we work with Rot-CIFAR10 dataset. This is because $E(2)$-CNNs are equivariant to larger groups than that of 90-degree rotations, so, we want to test the model's capabilities for these larger group symmetries. Here we build four variants of G-RepsCNNs: $T_1$-G-RepsCNN, $T_2$-G-RepsCNN, each for both $C_8$ and $C_{16}$ equivariance. Here $C_n$ for $n \in \{8, 16\}$ corresponds to the groups of $\frac{360}{n}$-degree rotations. The layer representations for $T_1$-G-RepsCNN are all $T_1$ tensors of the $C_n$ group. Whereas for $T_2$-G-RepsCNN, all layer representations except the last layer are $T_1$ representations and the last layer uses $T_2$ representations. Note that $T_1$-$E(2)$-CNN is the same as the traditional $E(2)$-CNN for the $C_n$ group, which has $T_1$ representation at each layer. $T_2$-G-Reps$E(2)$-CNN has $T_1$ representations for each layer except for the last layer that uses a $T_2$ representation. That is, the base architecture for constructing the G-Reps$E(2)$-CNN is $E(2)$-CNN with similar number of parameters and the $T_2$ layers are only used in the last layer. Please check Appendix D.2 for additional details on architecture.

For experiments on equivariant finetuning, we take the equituning algorithm of Basu et al. (2023b) that uses $T_1$ representations and extend it to use $T_2$ representations in the final layers. We use pretrained Resnet18 as

Table 6: The table shows mean (std) of percentage relative mean square errors over 3 seeds, and mean training time (in seconds) per epoch over 5 epochs and mean testing time per sample (in milliseconds) over 100 samples for solving PDEs with FNOs. G-RepsFNOs use regular $T_1$ representations with FNO as the base model. Both FNO and G-RepsFNO contain about 468K trainable parameters while G-FNO is much larger with about 3.4M parameters.

| Model / Dataset | FNO | | | G-RepsFNO | | | G-FNO | | |
|---|---|---|---|---|---|---|---|---|---|
| | MSE | Train time (s) | Test time (ms) | MSE | Train Time (s) | Test time (ms) | MSE | Train time (s) | Test time (ms) |
| NS | 8.41 (0.4) | 49.8 | 7.46 | 5.31 (0.2) | 53.9 | 7.71 | 4.78 (0.4) | 109.9 | 21.20 |
| NS-Sym | 4.21 (0.1) | 19.2 | 5.31 | 2.92 (0.1) | 20.8 | 7.52 | 2.24 (0.1) | 43.8 | 20.99 |

our non-equivariant base model and perform non-equivariant finetuning and equivariant finetuning with $T_1$ and $T_2$ representations. Additional experimental details are provided in Appendix D.

**Results:** In Tab. 3 and Tab. 5, we provide the results for training from scratch. From Tab. 3 and 5, we make two key observations: a) $T_2$-G-RepsCNNs are competitive and often outperform the baselines GCNNs and $E(2)$-CNNs, b) $T_2$ features, when added to the baselines to obtain $T_2$-G-RepsGCNNs and $T_2$-G-Reps$E(2)$CNNs, they outperform the original $T_1$ counterpart for both $C_8$ and $C_{16}$ equivariance. This shows the importance of higher-order tensors in image classification. Thus, we not only provide competitive performance to baselines using our models but also improve the results from these baselines by adding $T_2$ features in them. Finally, from Tab. 4, we find that on both rot90-CIFAR10 and Galaxy10, $T_2$-equitune easily outperforms equitune, confirming the importance of $T_2$ features.

## 5.4 Solving PDEs with FNOs

**Datasets and Experimental Setup:** We consider two versions of the incompressible Navier-Stokes equation from Helwig et al. (2023); Li et al. (2021). The first version is a Navier-Stokes equation without any symmetry (NS dataset) in the data, and a second version that does have 90° rotation symmetry (NS-SYM dataset). The general Navier-Stokes equation considered is written as,

$$\partial_t w(x,t) + u(x,t) \cdot \nabla w(x,t) = \nu \Delta w(x,t) + f(x), \tag{2}$$
$$\nabla \cdot u(x,t) = 0 \quad \text{and} \quad w(x,0) = w_0(x),$$

where $w(x,t) \in \mathbb{R}$ denotes the vorticity at point $(x,t)$, $w_0(x)$ is the initial velocity, $u(x,t) \in \mathbb{R}^2$ is the velocity at $(x,t)$, and $\nu = 10^{-4}$ is the viscosity coefficient. $f$ denotes an external force affecting the dynamics of the fluid. The task here is to predict the vorticity at all points on the domain $x \in [0,1]^2$ for some $t$, given the previous values of vorticity at all point on the domain for previous $T$ steps. As stated by Helwig et al. (2023), when $f$ is invariant with respect to 90° rotations, then the solution is equivariant, otherwise not. We use the same forces $f$ as Helwig et al. (2023). For non-invariant force, we use $f(x_1, x_2) = 0.1(\sin(2\pi(x_1 + x_2)) + \cos(2\pi(x_1 + x_2)))$ and as invariant force, we use $f_{inv} = 0.1(\cos(4\pi x_1) + \cos(4\pi x_2))$. We use $T = 20$ previous steps as inputs for the NS dataset and $T = 10$ for NS-SYM and predict for $t = T + 1$, same as in Helwig et al. (2023). We train our models with batch size 20 and learning rate $10^{-3}$ for 100 epochs.

**Model design:** We use the FNO and G-FNO models directly from Helwig et al. (2023). And we construct G-RepsFNO by directly using $T_1$ representation corresponding to the $C_4$ group of 90° rotations for all the features. Further, G-RepsFNO uses FNO as the base model.

**Results and Observations:** In Tab. 6, we find that G-RepsFNO clearly outperforms traditional FNOs on both datasets NS and NS-SYM. Note that the NS dataset does not have rot90 symmetries and yet G-RepsFNOs outperform FNOs showing that using equivariant representations may be more expressive for tasks without any obvious symmetries as was also noted in several works such as Cohen & Welling (2016a); Helwig et al. (2023). Moreover, we find that the G-RepsFNO models perform competitively with the more sophisticated, recently proposed, G-FNOs. Thus, we gain benefits of equivariance by directly using equivariant representations on non-equivariant base models and making minimal changes to the architecture. Further, in Tab. 6 we show that G-RepsFNOs are computationally much more efficient than the more sophisticated G-FNOs.

## 6  Limitations

Here we provide the limitations to our method to the best of our knowledge.

- Our method constructs lightweight networks equivariant to arbitrary matrix groups. However, the universality of our method is limited to orthogonal groups as discussed in §. 4.4. Universality for efficient practical equivariant architecture to general matrix groups is a challenging open problem and left for future research. Nevertheless, since orthogonal groups form a large class of groups of practical importance, we believe that the scalability of our network compared to equivariant networks as general as ours, e.g. Finzi et al. (2021b), Villar et al. (2021), makes it an important equivariant network construction method.

- While the G-RepsNet architecture is indeed general and applicable to arbitrary matrix groups, the function that is invariant to group action may not be easy to define or compute in general, unlike the case of orthogonal groups $O(n)$ and $O(1, n)$.

- Tensor operations used in our construction in Section 4 can be expensive for tensor of very high order. But in the majority of applications we use tensor multiplications only when the applications require such higher-order tensor features, e.g. in Section 5.1. In such cases, the applications themselves require higher-order features in the network to achieve good performance. E.g., the $O(3)$-equivariant regression task in Section 5.1 has second-order tensor outputs, hence, any equivariant network for this task must have higher-order tensors and hence, are computationally more expensive than non-equivariant networks. In second-order image classification in Section 5.3, we only use second-order tensor in the final few layers where it leads to significant improvement in performance and the feature dimensions are relatively not too large, hence, maintaining a comparable computational complexity (cf. Tab. 9, 10) to the equivariant networks using first-order features.

- The G-RepsNet architecture is designed to be simple and efficient (and universal for orthogonal groups) which means that we avoid tensor decomposition (e.g., required if $T_2$ tensors need to be converted to $T_1$). However, this leads to higher memory requirements. In cases with more extreme memory constraints, tensor decomposition may be worth despite the slower speed.

## 7  Conclusion

We present G-RepsNet, a lightweight yet expressive architecture designed to provide equivariance to arbitrary matrix groups. We find that G-RepsNet gives competitive performance to EMLP on various invariant and equivariant regression tasks taken from Finzi et al. (2021b), at much less computational expense. For image classification, we find that G-RepsNet with second-order tensors outperforms existing equivariant models such as GCNNs and $E(2)$-CNNs as well as methods such as equitune when trained using pretrained models such as Resnet. Further illustrating the simplicity and generality of our design, we show that using simple first-order tensor representations in G-RepsNet achieves competitive performance to specially designed equivariant networks for several different domains. We considered diverse domains such as PDE solving and $N$-body dynamics prediction using FNOs and MPNNs, respectively, as the base model.

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

# Appendix

# A  Additional Details on Related Works

## A.1  EMLPs and Universal Scalars

**EMLPs**   Given the input and output types for some matrix group, the corresponding tensor representations can be derived from the given base group representation $\rho$. Using these tensor representations, one can solve for the space of linear equivariant functions directly from the obtained equivariant constraints corresponding to the tensor representations. Finzi et al. (2021b) propose an elegant solution to solve these constraints by computing the basis of the linear equivariant space and construct an equivariant MLP (EMLP) from the computed basis. Our work is closest to this work as we use the same data representations as Finzi et al.

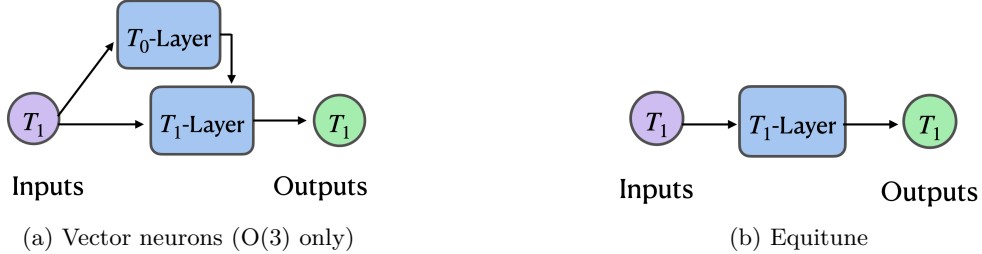

(a) Vector neurons (O(3) only)  (b) Equitune

Figure 6: (a) and (b) show layers from vector neurons Deng et al. (2021) and equitune Basu et al. (2023b), which are special cases of G-RepsNet..

(2021b), but we propose a much simpler architecture for equivariance to arbitrary matrix groups. Because of the simplicity of our approach, we are able to use it for several larger datasets, which is in contrast to Finzi et al. (2021b), where the experiments are mostly restricted to synthetic experiments. Moreover, using these bases are in general known to be computationally expensive (Fuchs et al., 2020).

**Universal scalars** Villar et al. (2021) propose a method to circumvent the need to explicitly use these equivariant bases. The First Fundamental Theorem of Invariant Theory for the Euclidean group $O(d)$ states that "a function of vector inputs returns an invariant scalar if and only if it can be written as a function only of the invariant scalar products of the input vectors" (Weyl, 1946). Taking inspiration from this theorem and a related theorem for equivariant vector functions, Villar et al. (2021) characterize the equivariant functions for various Euclidean and Non-Euclidean groups. They further motivate the construction of neural networks taking the invariant scalar products of given tensor data as inputs. However, the number of invariant scalars for $N$ tensors in a data point grows as $N^2$, hence, making it an impractical method for most real life machine learning datasets. Hence, their experiments are also mostly restricted to synthetic datasets like in EMLP.

Moreover, Villar et al. (2021, §. 5) show that even though the number of resulting scalars grows proportional to $N^2$, when the data is of dimension $d$, approximately $N \times (d+1)$ number of these scalars is sufficient to construct the invariant function. But, it might not be trivial to find this subset of scalar for real life datasets such as images. Hence, we propose to use deeper networks with equivariant features that directly take the $N$ tensors as input, instead of $N^2$ scalar inputs, which also circumvent the need to use equivariant bases. Additional related works and comparisons are in Section 2.

## A.2 Special Cases and Related Designs

Here, we look at existing group equivariant architectures popular for their simplicity that are special cases or closely related to our general design.

**Vector Neurons** Popular for its lightweight $SO(3)$-equivariant applications such as point cloud, the vector neurons (Deng et al., 2021) serve as a classic example of special cases of our design as illustrated in Fig. 6a. Their $T_1$-layer simply consists of a linear combination $T_1$ inputs without bias terms, same as ours. The $T_0$-layer first converts the $T_1$ tensors into $T_0$ tensors by taking inner products. Then, pointwise non-linearities are applied to the $T_0$ tensor and then mixed with the $T_1$ tensors, by multiplying them with $T_1$ tensors and further linearly mixing the $T_1$ tensors.

**Harmonic networks** Harmonic networks or H-nets (Worrall et al., 2017) employ a similar architecture to ours and vector neurons, but specialized for the $SO(2)$ group. They also take as input $T_1$ inputs, then obtain the $T_0$ scalars by computing the Euclidean norms of the inputs. All non-linearities are applied only to the scalars. The $T_1$ tensors are processed using linear circular cross-correlations that preserve equivariance. Further, higher order tensors are obtained by chained-cross correlations. The use of cross-correlations is very different from our design , but it is designed in a similar spirit of building tensors of various orders and construct simple, yet expressive equivariant features.

**Deepsets** Deepsets (Zaheer et al., 2017) is a popular architecture equivariant to permutations. Here, we show how Deepsets can be constructed using non-regular representation for the $S(n)$ group of permutations using the G-RepsNet construction in §. 4.

First, we recall that the Deepsets architecture. Suppose the input to the Deepsets is given by $\mathbf{X} = [X_1, X_2, X_3]$, then the output of each layer of Deepsets is given by $\mathbf{Y} = [Y_1, Y_2, Y_3]$, where $Y_i = W_1(X_i) + W_2(\sum_{i=1}^{3} X_i)$, where $W_1$ and $W_2$ are two learnable weight matrices. Usually, we also have pointwise activation functions used on the output, which we ignore here since they do not affect the equivariance of permutation groups. It is easy to verify that the Deepsets layer is equivariant to permutations on $\mathbf{X}$, since, if we replace $X_i$ by $X_j$ in the input for $i \neq j$, then $Y_i$ is replaced by $Y_j$ in the output.

Now, consider the construction of G-RepsNet corresponding to the $S(n)$ group using non-regular representation from Section 4. For the input $\mathbf{X} = [X_1, X_2, X_3]$, the group channel dimension is equal to 1. We first obtain the $T_0$ and $T_1$ representations of the data as $X_{T_0} = \sum_{i=1}^{3} X_i, X_{T_1} = [X_1, X_2, X_3]$. Then, we obtain $Y_{T_0} = [W_1(X_{T_0}), 1]$, which is a concatenation of the invariant term $W_1(X_{T_0})$ and a scalar 1 independent of the input. Similarly, obtain the $H_{T_1} = [1, W_2(X_{T_1})]$, where $W_2(X_{T_1}) := [W_2(X_1), W_2(X_2), W_2(X_3)]$. Now, we get $Y'_{T_1} = H_{T_1} * Y_{T_0} = [W_1(X_{T_0}), W_2(X_{T_1})]$. From $Y'_{T_1}$, it is easy to obtain $Y_{T_1}$ by summing the two components $W_1(X_{T_0})$ and $W_2(X_{T_1})$ of $Y'_{T_1}$. Thus, establishing the similarity between Deepsets and G-RepsNet for the permutation group. Note that here we used the $\mathrm{inv}(\cdot)$ as a constant, which is a valid choice, to obtain a similar form of the output as Deepsets.

**Equitune** Finally, recent works on frame-averaging such as equitune and related symmetrization techniques (Basu et al., 2023a; Kim et al., 2023; Kaba et al., 2023; Mondal et al., 2023) construct equivariant architecture by performing some sort of averaging over groups. This can be seen as using a regular $T_1$ representation as the input and output type as illustrated in Fig. 6b. These works have mainly focused on exploring the potential of equivariance in pretrained models. In this work, we further explore the capabilities of regular $T_1$ representations and find their surprising benefits in equivariant tasks. Moreover, this also inspires us to explore beyond regular $T_1$ representations, e.g., we find $T_2$ representations can yield better results than $T_1$ representations when used in the final layers of a model for image classification.

# B    Proof of Equivariance

Here we provide the proof of equivariance of a G-RepsNet layer to matrix groups. Further, since stacking equivariant layers preserve the equivariance of the resulting model, the equivariance of the G-RepsNet model follows directly. The argument is similar to the proof of equivariance of vector neurons to the $\mathrm{SO}(3)$ group.

First, consider regular representation. Note from Section 4 that the group channel dimension is treated like a batch dimension in regular representations for discrete groups. Thus, any permutation in the input naturally appears in the output, hence, producing equivariant output.

Now we consider non-regular representations. Assuming that the input to a G-RepsNet layer consists of tensors of types $T_0, T_1, \ldots, T_n$, we first note that the output of the $T_0$-layer in Fig. 1a is invariant, following which we find that the $T_i$-layer outputs equivariant $T_i$ tensors.

The output of the $T_0$-layer is clearly invariant since all the inputs to the network are of type $T_0$, which are already invariant.

Now, we focus on a $T_i$-layer. Recall from Section 4 that a $T_i$ layer consists only of linear networks without any bias terms or pointwise non-linearities. Suppose the linear network is given by a stack of linear matrices. We show that any such linear combination performed by a matrix preserves equivariance, hence, stacking these matrices would still preserve equivariance of the output. Let the input tensor of type $T_i$ be $X \in \mathbb{R}^{c \times k}$, i.e., we have $c$ tensors of type $T_i$ and size of the representation of each tensor equals to $k$. Consider a matrix $W \in \mathbb{R}^{c' \times c}$, which multiplied with $X$ gives $Y = W \times X \in \mathbb{R}^{c' \times k}$, where $Y$ is a linear combination of the $c$ input tensors each of type $T_i$. Let the group transformation on the tensor $T_i$ be given by $G \in \mathbb{R}^{k \times k}$. Then the group transformed input is given by $X' = X \times G \in \mathbb{R}^{c \times k}$. The output of $X'$ through the $T_i$-layer is given by $Y' = W \times X \times G \in \mathbb{R}^{c' \times k} = (W \times X) \times G = Y \times G$, where the second last equality follows from the associativity property of matrix multiplication. Thus, each $T_i$-layer is equivariant.

# C   On the Universality of the G-RepsNet Architecture

*Proof to Thm. 1.* Let the tensors of type $i$ at layer $l$ be written as $H_i^l$. Given input $(X_1, X_2, \ldots, X_n) \in \mathbb{R}^d$ of type $T_1$, we construct a G-RepsNet architecture that can approximate $h$ by taking help from the approximation properties of a multi-layered perceptron (Hornik et al., 1989).

Let the first layer consist only of $T_1$-layers, i.e., linear layers without any bias terms such that the obtained hidden layer $H_1^1$ is of dimension $\mathbb{R}^{d \times (n^2+n)}$ and consists of the $T_1$ tensors $X_i + X_j$ for all $i, j \in \{1, \ldots, n\}$ and $X_i$ for all $i \in \{1, \ldots, n\}$. This can be obtained by a simple linear combination. Now, construct the second layer by first taking the norm of all the $T_1$ tensors, which gives $\|X_i\| + \|X_j\| + 2 \times \langle X_i, X_j \rangle$ for all $i, j \in \{1, \ldots, n\}$ and $\|X_i\|$ for all $i \in \{1, \ldots, n\}$. Then, using a simple linear combination of the converted $T_0$ tensors give $\langle X_i, X_j \rangle$ for all $i, j \in \{1, \ldots, n\}$. Finally, passing $\langle X_i, X_j \rangle$ for all $i, j \in \{1, \ldots, n\}$ through an MLP gives $H_0^2$. Now, from the universal approximation capability of MLPs, it can approximate $f$ from equation 1. Thus, we obtain the function $h$ from Lem. 1.

$\square$

Now, recall from Villar et al. (2021), a statement similar to Lem. 1, but for vector functions.

**Lemma 2** (Villar et al. (2021))**.** *A function of vector inputs returns an equivariant vector if and only if it can be written as a linear combination of invariant scalar functions times the input vectors. That is, given input vectors $(X_1, X_2, \ldots, X_n)$, $X_i \in \mathbb{R}^d$, any equivariant vector function $h : \mathbb{R}^{d \times n} \mapsto \mathbb{R}^d$ can be written as*

$$h(X_1, X_2, \ldots, X_n) = \sum_{t=1}^{n} f_t(\langle X_i, X_j \rangle_{i,j=1}^n) X_t, \tag{3}$$

*where $\langle X_i, X_j \rangle$ denotes the inner product between $X_i$ and $X_j$, and $f_t$s are some arbitrary functions.*

*Proof to Thm. 2.* The proof closely follows the proof for Thm. 1. Let the tensors of type $i$ at layer $l$ be written as $H_i^l$. Given input $(X_1, X_2, \ldots, X_n) \in \mathbb{R}^d$ of type $T_1$, we construct a G-RepsNet architecture that can approximate $h$ by taking help from the approximation properties of a multi-layered perceptron Hornik et al. (1989).

Let the first layer consist only of $T_1$-layers, i.e., linear layers without any bias terms such that the obtained hidden layer $H_1^1$ is of dimension $\mathbb{R}^{d \times (n^2+n)}$ and consists of the $T_1$ tensors $X_i + X_j$ for all $i, j \in \{1, \ldots, n\}$ and $X_i$ for all $i \in \{1, \ldots, n\}$. This can be obtained by a simple linear combination.

Let the second layer consist of both a $T_0$ layer and a $T_1$ layer. Let the $T_0$ layer output, $H_0^1$, be $\langle X_i, X_j \rangle$ for all $i, j \in \{1, \ldots, n\}$ and $\|X_i\|$ for $i \in \{1, \ldots, n\}$ in a similar way as done in the proof for Thm. 1. And let the $T_1$ layer output, $H_1^1$, be $X_i$ for $i \in \{1, \ldots, n\}$.

Again, let the third layer also consist of a $T_0$ layer and a $T_1$ layer. Let the $T_0$ layer consist of MLPs approximating the output $\|X_t\| \times f_t(\langle X_i, X_j \rangle_{i,j=1}^n)$ for $t \in \{1, \ldots, n\}$. Denote $\|X_t\| \times f_t(\langle X_i, X_j \rangle_{i,j=1}^n)$ as $H_0^{3,t}$. Then, let the $T_1$ layer consist of first mixing the scalars $H_0^{3,t}$ with $X_t$ as described in Section 4 as

$$H_1^{3,t} = X_t \times \frac{H_0^{3,t}}{\|X_t\|},$$

where $H_1^{3,t}$ for $t \in \{1, \ldots, n\}$ represent the output of the $T_1$ layer of the third layer. Note that from the universal approximation properties of MLPs (Hornik et al., 1989), we get that $H_1^{3,t}$ approximates $X_t \times f_t(\langle X_i, X_j \rangle_{i,j=1}^n)$. Finally, the fourth layer consists of a single $T_1$ layer that sums the vectors $H_1^{3,t}$ for $t \in \{1, \ldots, n\}$, which combined with Lem. 2 concludes the proof.

$\square$

Thus, we find that a simple architecture can universally approximate invariant scalar and equivariant vector functions for the $O(d)$ or $O(1, d)$ groups. This is reminiscent of the universality property of a single-layered MLP. However, in practice, deep neural networks are known to have better representational capabilities than

Table 7: Test loss and running time for various neural network architectures for the 5-body dynamics prediction task. The results for SEGNN are taken from Brandstetter et al. (2022), rest are taken from Satorras et al. (2021).

| Model | Test Loss | Forward Time |
|---|---|---|
| Linear | 0.0819 | 0.0001 |
| SE (3) Transformer Fuchs et al. (2020) | 0.0244 | 0.1346 |
| Tensor Field Network Thomas et al. (2018) | 0.0155 | 0.0343 |
| Graph Neural Network Gilmer et al. (2017) | 0.0107 | 0.0032 |
| Radial Field Network Köhler et al. (2019) | 0.0104 | 0.0039 |
| EGNN Satorras et al. (2021) | 0.0071 | 0.0062 |
| SEGNN Brandstetter et al. (2022) | 0.0043 | 0.0260 |

a single-layered MLP. In a similar way, in practice, we design deep equivariant networks using the G-RepsNet architecture that provides good performance on a wide range of domains.

# D    Additional Experimental Details

## D.1    Comparison with EMLPs

Here we provide the learning rate and model sizes used for the experiments on comparison with EMLPs in Section 5.1.

For each task and model, we choose model sizes between small (with channel size 100) and large (with channel size 384). Similarly, we choose the learning rate from $\{10^{-3}, 3 \times 10^{-3}\}$.

In the $O(5)$-invariant regression task, for MLPs and EMLPs, we use a learning rate of $3 \times 10^{-3}$ and channel size 384. Whereas for G-RepsNets, we use a learning rate of $10^{-3}$ and channel size 100.

For the $O(3)$-equivariant task, we use learning rate $10^{-3}$ and channel size 384 for all the models.

For the $O(1,3)$-invariant regression task, we use a learning rate of $3 \times 10^{-3}$ for all the models. Further, we use a channel size of 384 for MLPs and EMLPs, whereas for G-RepsNets, a channel size of 100 was chosen as it gives better result.

## D.2    Second-Order Image Classification

**Training CNNs from scratch:** The CNN used for the ablation experiments for the plot in Fig. 4a consists of 3 convolutional layers each with kernel size 5, and output channel sizes 6, 16, and 120, respectively. Following the convolutional layers are 5 fully connected layers, each consisting of features of dimension of 120. For training from scratch, we train each model for 10 epochs, using stochastic gradient descent with learning rate of $10^{-3}$, momentum of 0.9. Further, we also use a stepLR learning rate scheduler with $\gamma$ of 0.1, step size of 7, which reduces the learning rate by a factor of $\gamma$ after every step size number of epochs.

**Comparison to GCNNs and $E(2)$-CNNs when trained from scratch:** For each of the models, we use the Resnet18 architecture (either with naive convolutions or group convolutions). We train each model for 100 epochs using stochastic gradient descent, with a learning rate $10^{-3}$, momentum of 0.9, weight decay of $5 \times 10^{-4}$.

**Second-Order Finetuning:** For finetuning the pretrained Resnet18, we use 5 epochs, using stochastic gradient descent with learning rate of $10^{-3}$, momentum of 0.9. For equivariant finetuning with $T_2$ representations, we first extract $T_1$ featured from the pretrained model same as done for equituning (Basu et al., 2023b), following which we convert it to $T_2$ representations using a simple outer product. Once the desired features are obtained, we pass it through two fully connected layers with a ReLU activation function in between to obtain the final classification output.

Table 9: Table shows mean training time (in minutes) taken by each of the models per epoch for the experiments in Tab. 3. We note that $T_2$-G-RepsCNN is computationally comparable to $T_1$-G-RepsCNN as the second-order features in $T_2$-G-RepsCNN are only used in the final layers with small dimensions. Similarly, GCNN is computationally comparable to $T_2$-G-RepsGCNN.

| Dataset \ Model | CNN | $T_1$-G-RepsCNN | $T_2$-G-RepsCNN | GCNN | $T_2$-G-RepsGCNN |
|---|---|---|---|---|---|
| Rot90-CIFAR10 | 1.07 | 3.19 | 3.21 | 5.50 | 5.50 |

Table 10: Table shows mean training time (in minutes) taken by each of the models per epoch for the experiments in Tab. 5. We note that $T_2$-G-RepsCNN is computationally comparable to $T_1$-G-RepsCNN as the second-order features in $T_2$-G-RepsCNN are only used in the final layers with small dimensions. Similarly, $E(2)$-CNN is computationally comparable to $T_2$-G-Reps$E(2)$CNN.

| Model | Equivariance | Tensor Orders | Training time (min) |
|---|---|---|---|
| CNN | – | – | 1.55 |
| $T_1$-G-RepsCNN | C8 | $(T_1)$ | 6.77 |
| $T_2$-G-RepsCNN | C8 | $(T_1, T_2)$ | 6.8 |
| $E(2)$-CNN | C8 | $(T_1)$ | 5.52 |
| $T_2$-G-Reps$E(2)$-CNN | C8 | $(T_1, T_2)$ | 5.52 |
| $T_1$-G-RepsCNN | C16 | $(T_1)$ | 9.80 |
| $T_2$-G-RepsCNN | C16 | $(T_1, T_2)$ | 9.17 |
| $E(2)$-CNN | C16 | $(T_1)$ | 6.81 |
| $T_2$-G-Reps$E(2)$-CNN | C16 | $(T_1, T_2)$ | 6.81 |

## E  Additional Results

### E.1  Comparison of time for forward passes in GNN models

We present the results of forward pass times for various equivariant and non-equivariant graph neural network models in Tab. 7 taken directly from Satorras et al. (2021). It shows that networks constructed from equivariant bases such as tensor field networks (TFNs) and $SE(3)$-equivariant transformers can be significantly slower than non-equivariant graph neural networks.

### E.2  Additional results on compute required for second-order image classification using GCNNs, $E(2)$-CNNs

Tab. 9 and 10 show the average training time taken per epoch for various models for the experimental results in Tab. 3 and 5, respectively.

