# OpenReview forum: "G-RepsNet: A Lightweight Construction of Equivariant Networks for Arbitrary Matrix Groups"
_TMLR — Accepted by TMLR_

### Review · Reviewer_wEwX · 2025-02-07

**Summary Of Contributions:**

The paper proposes GRepsNet, a way to construct equivariant networks.

As in prior work, features are collected in terms of which representation acts on them. The main idea is to, given a feature $X$ with associated representation/"feature type" $\rho:G\to\mathrm{GL}(\mathbb{R}^K)$, write the feature $X$ as a $K\times C$-tensor, where the representation dimension $K$ and "channel dimension" $C$ are kept apart. Then linear layers can be applied from the right, e.g. mapping $X\mapsto X W$, where $W$ is a learnable $C\times D$ weight matrix so that the output has new channel dimension $D$ and the same feature type as the input. The contribution of the present paper is to use this idea for quite general representations (when $\rho$ is the 3-dim irrep of $\mathrm{SO}(3)$, the idea is the well-known "Vector Neurons"-approach).

The paper proposes stacking a couple of linear layers as described above, interspersed with simple layers that mix feature types to build equivariant networks.

**Audience:**

Yes

**Claims And Evidence:**

No

**Requested Changes:**

# Critical changes for recommending acceptance
### On alignment with prior work and delineating the novelty of the work
1. For irreps (of real type), the proposed linear layers are general (due to Schur's lemma, see e.g. B.5.10 in [1]), but for other representations, the proposed linear layers do not represent maximally general linear layers. The present approach seems related to separable group convolutions as proposed in [2] and [3]. However, it is not intuitive to me why "separating out" higher tensor powers of the regular representation, and further not having any linear layer over the group axis, is a good idea.
2. Tensor products of regular representations decompose into direct sums of regular representations. For instance, $T_2$ features for the group of $90$ degree rotations decompose into four $T_1$'s. It is, therefore, unclear what the benefit of using $T_2$ features in the image classification experiment is. Is it the nonlinearity introduced by tensoring $T_1$ features? Is it the fact that we get four times as many features as input to the final classification layer? Or is it something else? Specifically, statements like "This shows the importance of higher-order tensors in image classification." at the end of Section 5.3, are therefore not evident.
3. The tensor products of representations considered as feature types in this work decompose into irreps. So, all considered network features can be described in terms of irreps. This makes all CNNs considered here special cases of steerable CNNs, and it would help the reader if they were described as such.
4. As mentioned earlier, it is not clear to me how the mapping to invariant features is done in the CNN experiments.
5. I believe that the layers merging $T_0$ and higher-order tensors could be seen as special cases of the bilinear layers by Finzi et al.

### Experimental details
1. Move the figures and tables referenced from the main text to the main text. E.g. Fig. 6a and Table 8.
2. Include runtime and the number of parameters for all models.
3. If $T_2$ outperforms $T_1$, why not use $T_3$, $T_4$ etc?

### Technical background
The comments refer to sentences in Section 3.
1. "Then, for a group G, the linear group representation of G is defined as" -> *a* linear group representation
2. "the size of regular representation is proportional to the size of G." -> is *equal* to the size of G.
3. "Hence, the size of any representation, m, can be written as m = |G| × d for some integer d." Not the size of any representation, but specifically the tensor product representations considered here.

# Non-critical changes, but highly recommended
1. Change to networks that achieve > 90% test accuracy in the CIFAR-10 experiments.

# References
[1] Weiler, Maurice, Patrick Forré, Erik Verlinde, and Max Welling. "Equivariant and Coordinate Independent Convolutional Networks." A Gauge Field Theory of Neural Networks (2023).

[2] Lengyel, Attila, and Jan van Gemert. "Exploiting learned symmetries in group equivariant convolutions." In 2021 IEEE International Conference on Image Processing (ICIP), pp. 759-763. IEEE, 2021.

[3] Knigge, David M., David W. Romero, and Erik J. Bekkers. "Exploiting redundancy: Separable group convolutional networks on lie groups." In International Conference on Machine Learning, pp. 11359-11386. PMLR, 2022.

**Strengths And Weaknesses:**

The main strength of the work is that building more efficient equivariant networks is important, as a common drawback of equivariant architectures is their high computational demand. The proposed networks seem simple, and this is a contribution as many equivariant networks are quite complicated.

### Weaknesses related to presentation
Since TMLR does not require novelty, there is no reason not to be charitable and relate as much as possible to prior work in the method description. This would improve the reader's experience significantly. Further, several tables referred to in the main text appear in the appendix, which is unnecessary as there is no page limit for TMLR. Please refer to the requested changes below for detailed comments.

### Experimental weaknesses
I will focus on the experiments on image data in Section 5.3, as these fall within my area of expertise.
1. The reported test accuracies for CIFAR-10 in Table 4 are very low. This immediately catches the eye of the reader and makes it difficult to assess the merits of the proposed method. For reference, in 2010, 79% was considered good (https://www.cs.toronto.edu/~kriz/conv-cifar10-aug2010.pdf) and in 2014, results above 80% were standard (https://code.google.com/archive/p/cuda-convnet/). Cohen & Welling got over 90% accuracy in 2016 with their G-CNNs. Thus, results below 80% (including for G-CNNs!) stand out to the reader. I do not see a reason not to have better architectures in these experiments. CIFAR-10 with rotations should not be more difficult than ordinary CIFAR-10 for rotation-invariant models.
2. I have trouble understanding exactly how $T_2$-features are mapped to $T_0$-features in the networks. Earlier in the paper, it is stated  that $T_0$-features are obtained through "an appropriate invariant operator, e.g. Euclidean norm for Euclidean groups, or averaging over the group channel dimension for regular groups." But I did not find information on what is used for the experiments in Section 5.3.
3. I did not find information on runtime or number of parameters for the different models in the experiments in Section 5.3.

---

> ### Author Response · Authors · 2025-03-13
> **Response to Reviewer wEwX. Thank you for your comments and suggestions**
>
> We thank the reviewer for their comments and suggestions. We address all their critical concerns below.
>
> “On alignment with prior work and delineating the novelty of the work”:
> 1. Our design is focused on generality as well as simplicity and efficiency, and is universal for a broad class of groups important for practical applications (orthogonal groups). We did not include any component – including linear layer over the group axis – that did not seem to contribute directly to the universality or empirically observed improved expressivity for these groups. That is, we start with tensors of various orders and add neural network processing between tensors of different orders where necessary. This adds to the simplicity of the design. We believe exploring the benefits of the linear layer over the group axis and understanding its impact, theoretically and practically, is an interesting question, and may help in some tasks.
>
> 2. We introduce the $T_2$ features in the CNN using outer products between the original $T_1$ features with the hypothesis that $T_2$ features, because they explicitly encode the second-order interactions, are able to capture more fine-grained details. Prior work like Bilinear CNNs (Lin et al., CVPR 2015) are also designed with this motivation. However, in our case the interactions are purely in the group dimension (in order to preserve equivariance), rather than across the spatial dimensions.
>
> 3. We thank the reviewer for suggesting the connection with steerable CNNs, we have added this point in Section 5.3 where we discuss the model design for G-RepsCNNs. We would like to note that, as also mentioned in EMLP (Finzi et al.), steerable CNNs require analytically solving for the filters given any group. We do not require any such analytical solution, significantly simplifying the design of equivariant networks for a broad class of functions, while also ensuring universality for orthogonal groups.
>
> 4. Computing the invariant from $T_2$ layers in G-RepsCNNs We simply take the mean of all elements in the matrix for mapping  $T_2$ to $T_0$. For example, for the $C_4$ group, if the T2 features are of dimension $16 \times d$, then the $T_0$ feature is obtained by taking the mean along the dimension of size $16$, thus giving the $T_0$ feature of dimension $d$. We have now mentioned this clearly in the paper.
>
> 5. The reviewer is correct in noting the similarity between the bilinear layer in EMLPs and the merging layers we use. The main difference is the motivation of the use of this component in the overall architecture: in EMLP the motivation is solely to enhance performance, whereas, in our design, it is motivated from the to make the design universal for orthogonal groups. Further, note that EMLPs do not provide any universality results. We have now mentioned this connection in the paper.
>
> Experimental details:
> 1. We agree with the reviewer’s suggestions and have moved the figure and table.
>
> 2. We have now included the runtime and number of parameters for the models in the appropriate sections in the paper.
>
> 3. As mentioned in the limitations section, as higher order tensors are computationally expensive, we only include them when the task necessitates it or when the performance improvement is substantial taking into account the higher complexity. We did not find practical benefit for tensor orders beyond $T_2$ in our experiments as none of the tasks required it and $T_3$ features can be expensive inside models like CNNs, especially with large feature dimensions.
>
> Technical background:
> 1. We have fixed all the typos. We thank the reviewer for finding them.

---

### Review · Reviewer_Pq7k · 2025-02-25

**Summary Of Contributions:**

This work introduces a novel framework for constructing efficient networks that are equivariant to arbitrary group transformations. Specifically, the authors propose a lightweight neural network architecture that utilizes tensor representations, which are processed and combined only using tensor addition and tensor multiplication operations. These simple operations preserve the equivariant properties of the network while requiring fewer computational resources compared to previous works, such as EMLP which constrains the linear operations between different equivariant representations by using a basis of equivariant weight matrices. After introducing the proposed architecture of Group Representation Networks (G-RepsNet), the authors show that it is a universal approximator in the case of orthogonal matrix groups. In the experimental section, they showcase how the proposed network desing can achieve performance comparable to the computationally expensive EMLP on a set of synthetic datasets. Additionally, they provide comparisons with other specialized equivariant architectures, showing improved performance in tasks such as image classification, Nbody simulation, and finding solutions for the 2D Navier-Stokes equation.

**Audience:**

Yes

**Broader Impact Concerns:**

There are no concerns on the ethical implications of the work that would require adding a Broader Impact Statement

**Claims And Evidence:**

Yes

**Requested Changes:**

Some possible changes that could further support the claims of this work are the following:
- A discussion about the assumption of the existence of an easy and efficient way to compute invariant functions in the case of arbitrary matrix groups. Is this computation always as easy as computing a vector norm?
- A discussion regarding the tradeoffs of using redundant information in higher-order tensor representations versus decomposing the input tensors into irreducibles. While the simplicity of the tensor representations provides benefits in the speed of the method, can it hurt the expressivity of the model given a fixed memory budget?
-  More experiments showcasing the scaling of the method when it uses tensors $T_i$ with $i\geq 3$, which could further support the scalability claims of the authors.

While the proposed changes are not necessary for acceptance, I believe they will help improve the presentation of the significance of this work.

**Strengths And Weaknesses:**

Strengths
- The simplicity of the proposed architecture allows it to be adapted to a wide range of equivariant architectures, as demonstrated in the experiments.
- The provided universality result indicates that the simplicity of the proposed architecture doesn't hurt its generality in the case of orthogonal groups. While this result is not generalized to arbitrary matrix groups, it is still quite significant, especially since the orthogonal groups cover the symmetries of a large portion of the commonly used equivariant tasks.
- The experimental results show that the proposed G-RespNets can achieve comparable results to the general EMLP in terms of performance while requiring significantly lower computational resources. Additionally, it compares favorably with other architectures specialized to equivariance in specific symmetry groups.

Weaknesses:
- One of the main weaknesses of the proposed approach is that the tensor product operation results in at least a quadratic increase in the dimension of the resulting representations. This can hurt the authors' scalability claims.
- Additionally, the choice not to perform tensor decomposition means that the higher-order tensors can contain correlated or redundant information. For example, a type-two tensor can contain a scalar part (trace), an antisymmetric part (vector), and a symmetric traceless part. While previous approaches that operate with irreducible representations can handle each of these parts separately, the proposed G-RespNet will need to process the information as a single entity with 9 correlated components. Could this redundancy necessitate larger latent representations to encode the same information compared to an approach that decomposes tensors into irreducibles?
- The design of the tensor mixing layer assumes the existence of both a scalar feature $Y_{T_0}$ and an invariant function inv(.). While in most groups with unitary representations this assumption is trivial and can be satisfied by using an appropriate norm, it can get more complex in the case of arbitrary matrix groups that are non-compact and do not have unitary representations. Is there a general solution for efficiently computing this invariant function inv(.) in the general case of arbitrary matrix groups?

---

> ### Author Response · Authors · 2025-03-13
> **Response to Reviewer Pq7k. Thank you for your comments and suggestions.**
>
> We thank the reviewer for their comments and suggestions. We address all their concerns below.
>
> 1. The reviewer correctly points out that the higher order tensors increase computational complexity, quickly in some cases. We agree and have discussed this in the Limitations section of the paper. Furthermore, our goal is to design a general and efficient architecture, where the inclusion of higher order representations is a design choice in many cases.. There are some cases like the O(3)-equivariant task where having $T_2$ representations is required as it is part of the task. Also, our architecture makes the placement of higher order tensors a design choice. We show that in the case of the image classification experiments that $T_2$ representations improve performance, and these representations are used only at the later stages of the network where the spatial dimensions of the feature are very small, thus not increasing the overall complexity significantly. Empirically, we find that, for this experiment, the forward time through the network essentially remains the same on a GPU.
>
> 2. In the general G-RepsNets architecture, whenever $T_2$ representations are present, the $T_1$ representations are also present. In the $O(3)$-equivariant inertia matrix prediction task, desired output is of type $T_2$ which necessitates the second-order representation. Furthermore, based on our experiments, we observe that having explicit $T_2$ representations in the case of CNNs improves performance, which we attribute to explicitly encoding second-order representations. Also, converting $T_2$ to $T_1$ requires tensor decomposition, which we avoid in our general architecture due to it being computationally slow.
>
> 3. It is true that finding an invariant function for an arbitrary group action is a non-trivial problem. For the various groups and domains of practical importance we find that in most practical cases, it is indeed easy to compute. Nevertheless, we agree with the reviewer that, theoretically, such a general function is difficult to find. We have now added this as a limitation of the method in the paper.
>
> 4. Based on our experiments, we find that tensor decomposition can be extremely slow, hence becoming the bottleneck. Nevertheless, we agree that in cases, where memory is extremely limited, then it may be worth using tensor decomposition despite their slow speed. However, this was not the case in the experiments we have conducted. We have added this point to the limitations section.
>
> 5. As mentioned in the limitations section, as higher order tensors are computationally expensive, we only include them when the task necessitates it or when the performance improvement is substantial taking into account the higher complexity. We did not find practical benefit for tensor orders beyond $T_2$ in our experiments as none of the tasks required it and $T_3$ features can be expensive inside models like CNNs, especially with large feature dimensions.

---

### Review · Reviewer_mbNm · 2025-02-27

**Summary Of Contributions:**

This paper introduces G-RepsNet, a new type of equivariant neural network that uses tensor representations and simple tensor operations to achieve generality, efficiency, and scalability. G-RepsNet is proven to be a universal approximator for functions equivariant to orthogonal groups, and it outperforms existing methods on various tasks, including image classification and N-body dynamics prediction, while being computationally efficient. This work provides a seemingly significant advancement in building equivariant neural networks, making them more practical for a wider range of applications.

**Audience:**

Yes

**Broader Impact Concerns:**

No broader impact statement included: I don't think one is necessary.

**Claims And Evidence:**

Yes

**Requested Changes:**

Please address the weaknesses noted above.

Avoid putting claims and analysis in the caption of tables and figures. For example, any statement with the word “outperforms” should be in the main text, rather than the caption. I would shorten the caption for Table 2 to “Table 2: Mean (std) test accuracies for equituning using a pretrained Resnet with Rot90-CIFAR10 and Galaxy10.” and move the other statements to the main text. Please apply the same to all captions.

Please use `\citep{}` for parenthetical citations and `\citet{}` for textual citations. Many of your citations, as currently written, should be parenthetical rather than textual, and some vice versa. For example, " equivariant models such as GCNNs Cohen & Welling (2016a) and E(2)-CNNs Weiler & Cesa (2019)." -> " equivariant models such as GCNNs (Cohen & Welling, 2016a) and E(2)-CNNs (Weiler & Cesa, 2019)." and “such as (Fuchs et al., 2020; Thomas et al., 2018)” -> “such as Fuchs et al. (2020) and Thomas et al. (2018)”

I like the brief background given by the Section 3 “Group and Representation Theory” to help a wider audience understand the rest of the work. To follow this level of definition, the notation $O(n)$ for orthogonal groups should be explicitly defined. Other group notation definitions are also quite scattered: there’s a few in Section 4.1 “Input Feature Representations”, so I would recommend adding a paragraph to the “Group and Representation Theory” section to consolidate them to one place.

Can Tables 4 and 11 be merged?

Generally, there are quite a few typos and errors. Please proofread your work. Here are some fixes.

Abstract: “MLPs (EMLPs). But this method does not scale well and” -> “MLPs (EMLPs), but this method does not scale well, and” Generally, do not begin sentences with “but”.

Sec. 1 Introduction, paragraph 1: “can be challenging both because they require” -> “can be challenging because they both require”

Sec. 1 Introduction, paragraph 3: “.as is noted by the authors eml.”: fix this citation.

Sec. 1 Introduction, paragraph 4: “computationally expensive Further” -> “computationally expensive. Further”

Sec. 1 Introduction, enumerated list: “which is are a class” -> “which are a class”

Sec. 2 Related Works, paragraph 2 Steerable networks: “are provided in Sec. A.” -> “are provided in Appendix A.” Using the word “Appendix” helps clarify that this section is found at the end of the paper.

Sec. 4.3 Neural Processing, final paragraph: “groups (see C make use” -> “groups (see Appendix C) make use”

Sec. 4.4 Properties, paragraph 2: “note that restricting G-RespNet to” -> “note that restricting G-RepsNet to”

Figure 2 caption: the caption text overlaps with the subfigure labels.

Sec. 5.1 Comparison with EMLPs, paragraph 2 Model design: “with 5 layers where use different tensor” -> “with 5 layers, using different tensor”

Appendix E.4. Table 9: “GRpesFNOs”->”GRepsFNOs”

Entire paper: be consistent about whether there is a dash in “G-RepsNet” or not. A quick search shows 123 occurrences with the dash and 22 without (not including in figure image files).

**Strengths And Weaknesses:**

This work presents a novel approach to constructing equivariant networks using operations on tensor representations. It generalizes multiple related works into a unified framework while showing good performance and efficiency. It is applicable to arbitrary symmetry groups (provided a tensor representation) and exhibits universality for orthogonal groups, making it broadly applicable to various domains and tasks with different symmetries.

The breadth of experiments is appreciated, but each one seems to compare the proposed architecture(s) to a single related work and sometimes a non-equivariant baseline. Are there any other relevant architectures to compare against for each experiment?

Each of the “model design” sections are quite heavy to read in paragraph form, making them lack clarity. I would recommend instead presenting each in pseudocode, then using more natural language to describe them at a higher level. Including a codebase in addition to this change would further clarify exactly how your methods work.

Generally, the comparisons of G-RepsNet to related works is useful to show how it generalizes many of them. However, some relationships are not clear, partially due to the lack of clarity already mentioned as well as some missing connections. Specifically, how does your proposed invariant gating (Sec. 4.3 Neural Processing, 4th paragraph) compare to the Gated Nonlinearities proposed in Weiler et al. (2018) and then used in EMLPs (Finzi et al., 2021)? Also, tG-RepsNet seems more similar to vector neurons than to EMLPs, yet the latter seems to take precedence in multiple sections of the paper.

To strengthen your paper, you should include experimentation for groups beyond orthogonal groups in the non-regular representation case. Even if related works don’t provide such datasets, you can find, collect, or generate your own datasets to show the breadth of your proposed methods. The permutation symmetry group is an obvious candidate that at least Zaheer et al. (2017) has already provided datasets and experiments for.

---

> ### Author Response · Authors · 2025-03-13
> **Response to Reviewer mbNm. Thank you for your comments and suggestions.**
>
> We thank the reviewer for their comments and suggestions. We address all their concerns below.
>
> 1. For the experiments, while there may be more equivariant networks we can compare to, we chose high-performing representative specialized equivariant models for each of the diverse tasks – such as EMLP, G-FNOs, EGNN – and compared performance as well as efficiency. Please note the goal of our experiments is to primarily demonstrate the simplicity as well as generality of our method, not to outperform specialized architectures..
>
> 2. Our invariant vs. gated non-linearities: The reviewer is correct in noting the similarity between $H_{T_i}*Y_{T_0}$ and gated non-linearity in EMLPs. The main difference lies in the motivation of the use of this component in the overall architecture: in EMLP the motivation is to enhance the performance of EMLPs, whereas, in our design, it is motivated from the to make the design universal for orthogonal groups as shown in Appendix C.
>
> 3. Comparison to Vector Neurons: It is true that G-RepsNet has similarities with Vector Neurons in that they are both efficient equivariant models based on having very simple linear operations over representations that preserve equivariance. G-RepsNet is more general than Vector Neurons, as it includes higher order representations, applicable to arbitrary matrix groups. Furthermore, we show that G-RepsNets are universal for orthogonal groups. We do not have experiments to compare G-RepsNets to Vector Neurons, as Vector Neurons are essentially a special case of G-RepsNets as shown in A.2. Moreover, as mentioned, Vector Neurons are designed for $T_1$ tensors and originally only for the O(3) group. We are more interested in EMLPs because EMLPs are general and work for arbitrary matrix groups, hence, the goal of our work is to show that the simple, general, and efficient design of G-RepsNet can achieve performance comparable to EMLPs as well as other specialized architectures in various domains.
>
> 4. Experiments beyond orthogonal groups: We appreciate the reviewer's suggestion, but it has been difficult to find standard datasets with practical importance for non-orthogonal groups in this work. For example, we compare with all the experiments provided in EMLP, which are all orthogonal groups (the Lorentz group is also non-compact). For the case of the permutation group, our design reduces to the Deepsets architecture as pointed out in appendix Appendix A.2. We hope this paper will lead to work in this direction in the future.
>
> 5. We have removed all the claims and analysis parts from captions of figures and tables, as the reviewer suggested.
>
> 6. We have fixed all the citations and typos in the paper. We thank the reviewer for finding them. We have merged the tables as suggested.
>
> 7. As suggested by the reviewer, we have added a paragraph on definitions of $O(n)$ and $O(1,n)$ groups. For the definitions of tensor representations and regular representations, they immediately follow the other background material, and we believe is better to be described in Section 4.1.
>
> 8. We have improved the readability of model design in Section 5.1. We will release the code for the GRepsNet architectures after paper acceptance.

---

### Author Response · Authors · 2025-03-13
**Overall author response**

We thank all the reviewers for their thoughtful comments and suggestions. We have individually addressed all the concerns below. We have also revised the manuscript with all the required changes that the reviewers suggested.

---

### Decision · Action_Editor_tbT6 · 2025-04-13

**Recommendation:** Accept as is

**Comment:**

There is a consensus in reviewers that this paper introduces an interesting method and deserves to be accepted.

**Audience:**

The paper is relevant to the TMLR community and I do believe that both the symmetries, the graph networks and tensor representations are interesting subjects in the community.

**Claims And Evidence:**

The paper introduces Group Representation Networks (G-RepsNets), a lightweight equivariant network designed for arbitrary matrix groups using tensor polynomials. G-RepsNets demonstrate competitive performance against existing models, while being computationally efficient.

The reviewers have raised some concerns over the practicality of the existing method and while the discussion seems to alleviated some of the concerns, the reviewers are still hesitant about the scalability of the method. Nevertheless, they do agree that this is interesting and recommend acceptance.